# Like-minded sources on Facebook are prevalent but not polarizing

Brendan Nyhan[1,25 ✉], Jaime Settle[2,25], Emily Thorson[3,25], Magdalena Wojcieszak[4,5,25], Pablo Barberá[6,25], Annie Y. Chen[7], Hunt Allcott[8], Taylor Brown[6], Adriana Crespo-Tenorio[6], Drew Dimmery[6,24], Deen Freelon[9], Matthew Gentzkow[10], Sandra González-Bailón[9], Andrew M. Guess[11,12], Edward Kennedy[13], Young Mie Kim[14], David Lazer[15], Neil Malhotra[16], Devra Moehler[6], Jennifer Pan[17], Daniel Robert Thomas[6], Rebekah Tromble[18,19], Carlos Velasco Rivera[6], Arjun Wilkins[6], Beixian Xiong[6], Chad Kiewiet de Jonge[6,26], Annie Franco[6,26], Winter Mason[6,26], Natalie Jomini Stroud[20,21,26] & Joshua A. Tucker[22,23,26]

Many critics raise concerns about the prevalence of 'echo chambers' on social media and their potential role in increasing political polarization. However, the lack of available data and the challenges of conducting large-scale field experiments have made it difficult to assess the scope of the problem[1,2]. Here we present data from 2020 for the entire population of active adult Facebook users in the USA showing that content from 'like-minded' sources constitutes the majority of what people see on the platform, although political information and news represent only a small fraction of these exposures. To evaluate a potential response to concerns about the effects of echo chambers, we conducted a multi-wave field experiment on Facebook among 23,377 users for whom we reduced exposure to content from like-minded sources during the 2020 US presidential election by about one-third. We found that the intervention increased their exposure to content from cross-cutting sources and decreased exposure to uncivil language, but had no measurable effects on eight preregistered attitudinal measures such as affective polarization, ideological extremity, candidate evaluations and belief in false claims. These precisely estimated results suggest that although exposure to content from like-minded sources on social media is common, reducing its prevalence during the 2020 US presidential election did not correspondingly reduce polarization in beliefs or attitudes.

Increased partisan polarization and hostility are often blamed on online echo chambers on social media[3–7], a concern that has grown since the 2016 US presidential election[8–10]. Platforms such as Facebook are thought to fuel extremity by repeatedly showing people congenial content from like-minded sources and limiting exposure to counterarguments that could promote moderation and tolerance[11–13]. Similarly, identity-reinforcing communication on social media could strengthen negative attitudes toward outgroups and bolster attachments to ingroups[14].

To assess how often people are exposed to congenial content on social media, we use data from all active adult Facebook users in the USA to analyse how much of what they see on the platform is from sources that we categorize as sharing their political leanings (which we refer to as content from like-minded sources; see Methods, 'Experimental design'). With a subset of consenting participants, we then evaluate a potential response to concerns about the effects of echo chambers by conducting a large-scale field experiment reducing exposure to content from like-minded sources on Facebook. This research addresses three major gaps in our understanding of the prevalence and effects of exposure to congenial content on social media.

First, we have no systematic measures of content exposure on platforms such as Facebook, which are largely inaccessible to researchers[2]. Web traffic data suggest that relatively few Americans have heavily skewed information diets[15–18], but less is known about what they see on

[1]Department of Government, Dartmouth College, Hanover, NH, USA. [2]Department of Government and Data Science, William and Mary, Williamsburg, VA, USA. [3]Maxwell School of Citizenship and Public Affairs, Syracuse University, Syracuse, NY, USA. [4]Department of Communication, University of California, Davis, CA, USA. [5]Amsterdam School of Communication Research, University of Amsterdam, Amsterdam, The Netherlands. [6]Meta, Menlo Park, CA, USA. [7]CUNY Institute for State and Local Governance, New York, NY, USA. [8]Environmental and Energy Policy Analysis Center, Stanford University, Stanford, CA, USA. [9]Annenberg School for Communication, University of Pennsylvania, Philadelphia, PA, USA. [10]Department of Economics, Stanford University, Stanford, CA, USA. [11]Department of Politics, Princeton University, Princeton, NJ, USA. [12]School of Public and International Affairs, Princeton University, Princeton, NJ, USA. [13]Department of Statistics and Data Science, Carnegie Mellon University, Pittsburgh, PA, USA. [14]School of Journalism and Mass Communication, University of Wisconsin-Madison, Madison, WI, USA. [15]Network Science Institute, Northeastern University, Boston, MA, USA. [16]Graduate School of Business, Stanford University, Stanford, CA, USA. [17]Department of Communication, Stanford University, Stanford, CA, USA. [18]School of Media and Public Affairs, The George Washington University, Washington, DC, USA. [19]Institute for Data, Democracy, and Politics, The George Washington University, Washington, DC, USA. [20]Moody College of Communication, University of Texas at Austin, Austin, TX, USA. [21]Center for Media Engagement, University of Texas at Austin, Austin, TX, USA. [22]Wilf Family Department of Politics, New York University, New York, NY, USA. [23]Center for Social Media and Politics, New York University, New York, NY, USA. [24]Present address: Research Network Data Science, University of Vienna, Vienna, Austria. [25]These authors contributed equally: Brendan Nyhan, Jaime Settle, Emily Thorson, Magdalena Wojcieszak, Pablo Barberá. [26]These authors jointly supervised this work: Chad Kiewiet de Jonge, Annie Franco, Winter Mason, Natalie Jomini Stroud, Joshua A. Tucker. ✉e-mail: nyhan@dartmouth.edu

social media. Prior observational studies of information exposure on platforms focus on Twitter, which is used by only 23% of the public[19–22], or the news diet of the small minority of active adult users in the US who self-identified as conservative or liberal on Facebook in 2014–2015[23]. Without access to behavioural measures of exposure, studies must instead rely on survey self-reports that are prone to measurement error[24,25].

Second, although surveys find associations between holding polarized attitudes and reported consumption of like-minded news[26,27], few studies provide causal evidence that consuming like-minded content leads to lasting polarization. These observed correlations may be spurious given that the people with extreme political views are more likely to consume like-minded content[28,29]. In addition, although like-minded information can polarize[30–32], most experimental tests of theories about potential echo chamber effects are brief and use simulated content, making it difficult to know whether these findings generalize to real-world environments. Previous experimental work also raises questions about whether such polarizing effects are common[18,33], how quickly they might decay[18,33], and whether they are concentrated among people who avoid news and political content[28].

Finally, reducing exposure to like-minded content may not lead to a corresponding increase in exposure to content from sources with different political leanings (which we refer to as cross-cutting) and could also have unintended consequences. Social media feeds are typically limited to content from accounts that users already follow, which include few that are cross-cutting and many that are non-political[22]. As a result, reducing exposure to like-minded sources may increase the prevalence of content from sources that are politically neutral rather than uncongenial. Furthermore, if content from like-minded sources is systematically different (such as in its tone or topic), reducing exposure to such content may also have other effects on the composition of social media feeds. Reducing exposure to like-minded content could also induce people to seek out such information elsewhere online (that is, not on Facebook[34]).

In this study, we measure the prevalence of exposure to content from politically like-minded sources among active adult Facebook users in the US. We then report the results of an experiment estimating the effects of reducing exposure to content from politically like-minded friends, Pages and groups among consenting Facebook users ($n$ = 23,377) for three months (24 September to 23 December 2020). By combining on-platform behavioural data from Facebook with survey measures of attitudes collected before and after the 2020 US presidential election, we can determine how reducing exposure to content from like-minded sources changes the information people see and engage with on the platform, as well as test the effects over time of reducing exposure to these sources on users' beliefs and attitudes.

This project is part of the US 2020 Facebook and Instagram Election Study. Although both Meta researchers and academics were part of the research team, the lead academic authors had final say on the analysis plan, collaborated with Meta researchers on the code implementing the analysis plan, and had control rights over data analysis decisions and the manuscript text. Under the terms of the collaboration, Meta could not block any results from being published. The academics were not financially compensated and the analysis plan was preregistered prior to data availability (https://osf.io/3sjy2); further details are provided in Supplementary Information, section 4.8.

We report several key results. First, the majority of the content that active adult Facebook users in the US see comes from like-minded friends, Pages and groups, although only small fractions of this content are categorized as news or are explicitly about politics. Second, we find that an experimental intervention reducing exposure to content from like-minded sources by about a third reduces total engagement with that content and decreases exposure to content classified as uncivil and content from sources that repeatedly post misinformation. However, the intervention only modestly increases exposure to content

from cross-cutting sources. We instead observe a greater increase in exposure to content from sources that are neither like-minded nor cross-cutting. Moreover, although total engagement with content from like-minded sources decreased, the rate of engagement with it increased (that is, the probability of engaging with the content from like-minded sources that participants did see was higher).

Furthermore, despite reducing exposure to content from like-minded sources by approximately one-third over a period of weeks, we find no measurable effects on 8 preregistered attitudinal measures, such as ideological extremity and consistency, party-congenial attitudes and evaluations, and affective polarization. We can confidently rule out effects of ±0.12 s.d. or more on each of these outcomes. These precisely estimated effects do not vary significantly by respondents' political ideology (direction or extremity), political sophistication, digital literacy or pre-treatment exposure to content that is political or from like-minded sources.

## Exposure to like-minded sources

Our analysis of platform exposure and behaviour considers the population of US adult Facebook users (aged 18 years and over). We focus primarily on those who use the platform at least once per month, who we call monthly active users. Aggregated usage levels are measured for the subset of US adults who accessed Facebook at least once in the 30 days preceding 17 August 2020 (see Supplementary Information, section 4.9.4 for details). During the third and fourth quarters of 2020, which encompass this interval as well as the study period for the experiment reported below, 231 million users accessed Facebook every month in the USA.

We used an internal Facebook classifier to estimate the political leaning of US adult Facebook users (see Supplementary Information, section 2.1 for validation and section 1.3 for classifier details; Extended Data Fig. 1 shows the distribution of predicted ideology score by self-reported ideology, party identification and approval of former president Donald Trump). The classifier produces predictions at the user level ranging from 0 (left-leaning) to 1 (right-leaning). Users with predicted values greater than 0.5 were classified as conservative and otherwise classified as liberal, enabling us to analyse the full population of US active adult Facebook users. A Page's score is the mean score of the users who follow the Page and/or share its content; a group's score is the mean score of group members and/or users who share its content. We classified friends, Pages or groups as liberal if their predicted value was 0.4 or below and conservative if it was 0.6 or above. This approach allows us to identify sources that are clearly like-minded or cross-cutting with respect to users (friends, Pages and groups with values between 0.4 and 0.6 were treated as neither like-minded nor cross-cutting).

We begin by assessing the extent to which US Facebook users are exposed to content from politically like-minded users, Pages and groups in their Feed during the period 26 June to 23 September 2020 (see Supplementary Information, section 4.2, for measurement details). We present estimates of these quantities among US adults who logged onto Facebook at least once in the 30 days preceding 17 August 2020.

We find that the median Facebook user received a majority of their content from like-minded sources—50.4% versus 14.7% from cross-cutting sources (the remainder are from friends, Pages and groups that we classify as neither like-minded nor cross-cutting). Like-minded exposure was similar for content classified as 'civic' (that is, political) or news (see Supplementary Information, section 4.3 for details on the classifiers used in this study). The median user received 55% of their exposures to civic content and 47% of their exposures to news content from like-minded sources (see Extended Data Table 1 for exact numbers and Supplementary Fig. 3 for a comparison with our experimental participants). Civic and news content make up a relatively small share of what people see on Facebook, however (medians of 6.9% and 6.7%, respectively; Supplementary Table 11).

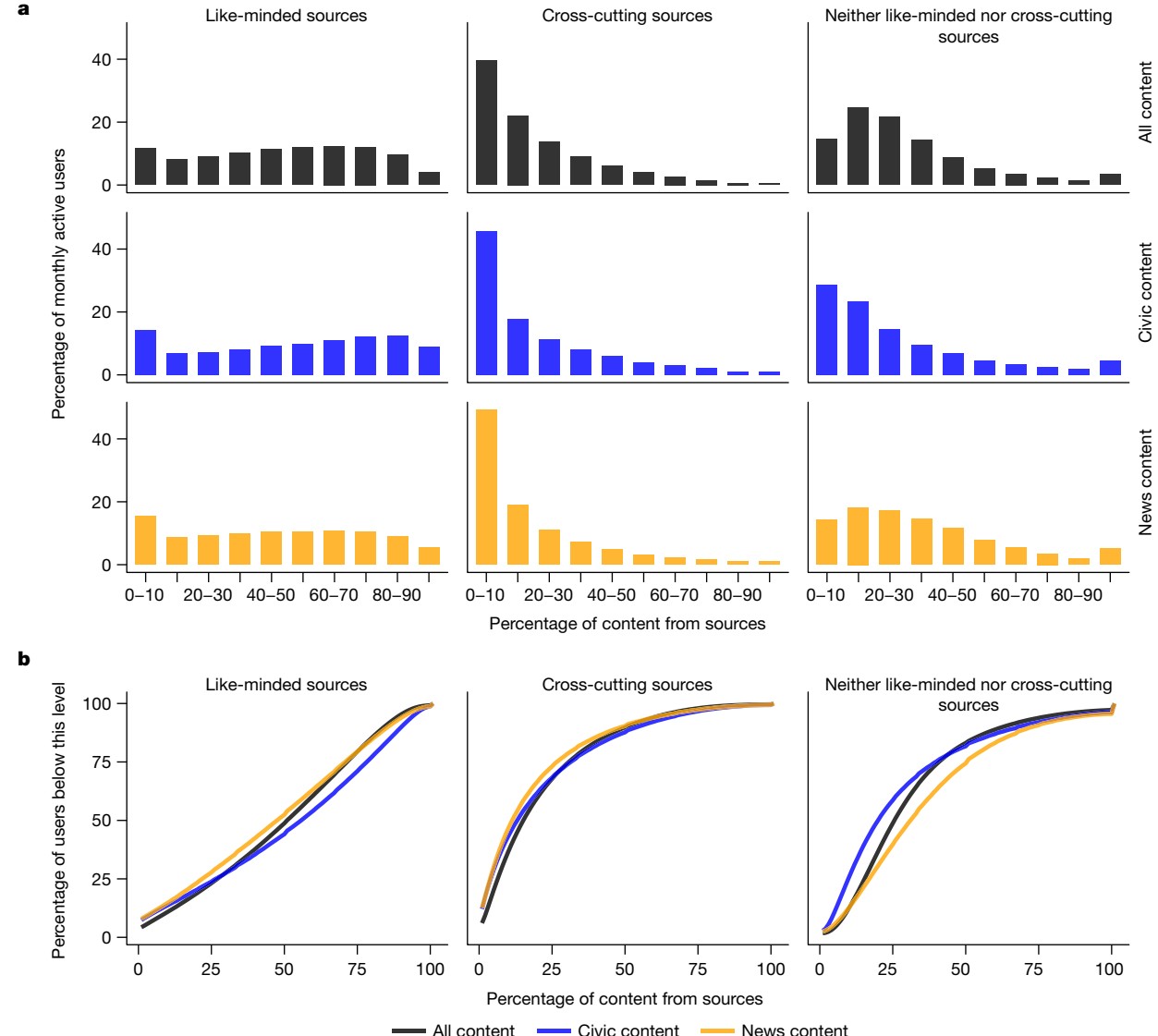

**Fig. 1 | The distribution of exposure to content among Facebook users.**
**a**, The distribution of the exposure of monthly active adult Facebook users in the USA to content from like-minded sources, cross-cutting sources, and those that fall into neither category in their Facebook Feed. Estimates are presented for all content, content classified as civic (that is, political) and news. **b**, Cumulative distribution functions of exposure levels by source type. Source and content classifications were created using internal Facebook classifiers (Supplementary Information, section 1.3).

However, patterns of exposure can vary substantially between users. Figure 1 provides the distribution of exposure to sources that were like-minded, cross-cutting or neither for all content, civic content and news content for Facebook users.

Despite the prevalence of like-minded sources in what people see on Facebook, extreme echo chamber patterns of exposure are infrequent. Just 20.6% of Facebook users get over 75% of their exposures from like-minded sources. Another 30.6% get 50–75% of their exposures on Facebook from like-minded sources. Finally, 25.6% get 25–50% of their exposures from like-minded sources and 23.1% get 0–25% of their exposures from like-minded sources. These proportions are similar for the subsets of civic and news content (Extended Data Table 1). For instance, like-minded sources are responsible for more than 75% of exposures to these types of content for 29% and 20.6% of users, respectively.

However, exposure to content from cross-cutting sources is also relatively rare among Facebook users. Only 32.2% have a quarter or more of their Facebook Feed exposures coming from cross-cutting sources (31.7% and 26.9%, respectively, for civic and news content).

These patterns of exposure are similar for the most active Facebook users, a group that might be expected to consume content from congenial sources more frequently than other groups. Among US adults who used Facebook at least once each day in the 30 days preceding 17 August 2020, 53% of viewed content was from like-minded sources versus 14% for cross-cutting sources, but only 21.1% received more than 75% of their exposures from like-minded sources (see Extended Data Fig. 2 and Extended Data Table 2).

These results are not consistent with the worst fears about echo chambers. Even among those who are most active on the platform, only a minority of Facebook users are exposed to very high levels of content from like-minded sources. However, the data clearly indicate that Facebook users are much more likely to see content from like-minded sources than they are to see content from cross-cutting sources.

### Experiment reducing like-minded source exposure
To examine the effects of reducing exposure to information from like-minded sources, we conducted a field experiment among consenting US adult Facebook users. This study combines data on participant

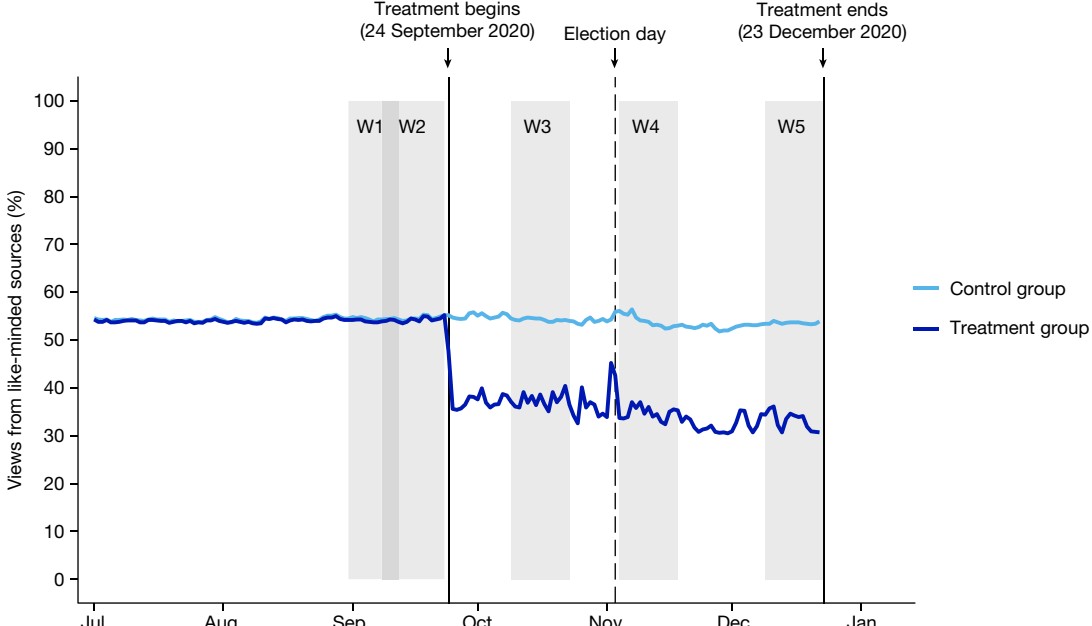

**Fig. 2 | Day-level exposure to content from like-minded sources in the Facebook Feed by experimental group.** Mean day-level share of respondent views of content from like-minded sources by experimental group between 1 July and 23 December 2020. Sources are classified as like-minded on the basis of estimates from an internal Facebook classifier at the individual level for users and friends, and at the audience level for Pages and groups. W1–W5 indicate survey waves 1 to 5; shading indicates wave duration. Extended Data Fig. 3 provides a comparable graph of views of content from cross-cutting sources. Note: exposure levels increased briefly on 2 and 3 November owing to a technical problem; details are provided in Supplementary Information, section 4.11.

behaviour on Facebook with their responses to a multi-wave survey, a design that allows us to estimate the effects of the treatment on the information that participants saw, their on-platform behaviour and their political attitudes (Methods).

Participants in the treatment and control groups were invited to complete five surveys before and after the 2020 presidential election assessing their political attitudes and behaviours. Two surveys were fielded pre-treatment: wave 1 (31 August to 12 September) and wave 2 (8 September to 23 September). The treatment ran from 24 September to 23 December. During the treatment period, 3 more surveys were administered: wave 3 (9 October to 23 October), wave 4 (4 November to 18 November) and wave 5 (9 December to 23 December). All covariates were measured in waves 1 and 2 and all survey outcomes were measured after the election while treatment was still ongoing (that is, in waves 4 and/or 5). Throughout the experiment, we also collected data on participant content exposure and engagement on Facebook.

In total, the sample for this study consists of 23,377 US-based adult Facebook users who were recruited via survey invitations placed at the top of their Facebook feeds in August and September 2020, provided informed consent to participate and completed at least one post-election survey wave (see Supplementary Information, sections 4.5 and 4.9).

For participants assigned to treatment, we downranked all content (including, but not limited to, civic and news content) from friends, groups and Pages that were predicted to share the participant's political leaning (for example, all content from conservative friends and groups and Pages with conservative audiences was downranked for participants classified as conservative; see Supplementary Information, section 1.1).

We note three important features of the design of the intervention. First, the sole objective of the intervention was to reduce exposure to content from like-minded sources. It was not designed to directly alter any other aspect of the participants' feeds. Content from like-minded sources was downranked using the largest possible demotion

strength that a pre-test demonstrated would reduce exposure without making the Feed nearly empty for some users, which would have interfered with usability and thus confounded our results; see Supplementary Information, section 1.1. Second, our treatment limited exposure to all content from like-minded sources, not just news and political information. Because social media platforms blur social and political identities, even content that is not explicitly about politics can still communicate relevant cues[14,35]. Also, because politics and news account for a small fraction of people's online information diets[18,36,37], restricting the intervention to political and/or news content would yield minimal changes to some people's Feeds. Third, given the associations between polarized attitudes and exposure to politically congenial content that have been found in prior research, we deliberately designed an intervention that reduces rather than increases exposure to content from like-minded sources to minimize ethical concerns.

### Treatment effects on content exposure

The observed effects of the treatment on exposure to content from like-minded sources among participants are plotted in Fig. 2. As intended, the treatment substantially reduced exposure to content from like-minded sources relative to the pre-treatment period. During the treatment period of 24 September to 23 December 2020, average exposure to content from like-minded sources declined to 36.2% in the treatment group while remaining stable at 53.7% in the control group ($P < 0.01$). Exposure levels were relatively stable during the treatment period in both groups, except for a brief increase in treatment group exposure to content from like-minded sources on 2 November and 3 November, owing to a technical problem in the production servers that implemented the treatment (see Supplementary Information, section 4.11 for details).

Our core findings are visualized in Fig. 3, which shows the effects of the treatment on exposure to different types of content during the treatment period (Fig. 3a), the total number of actions engaging with

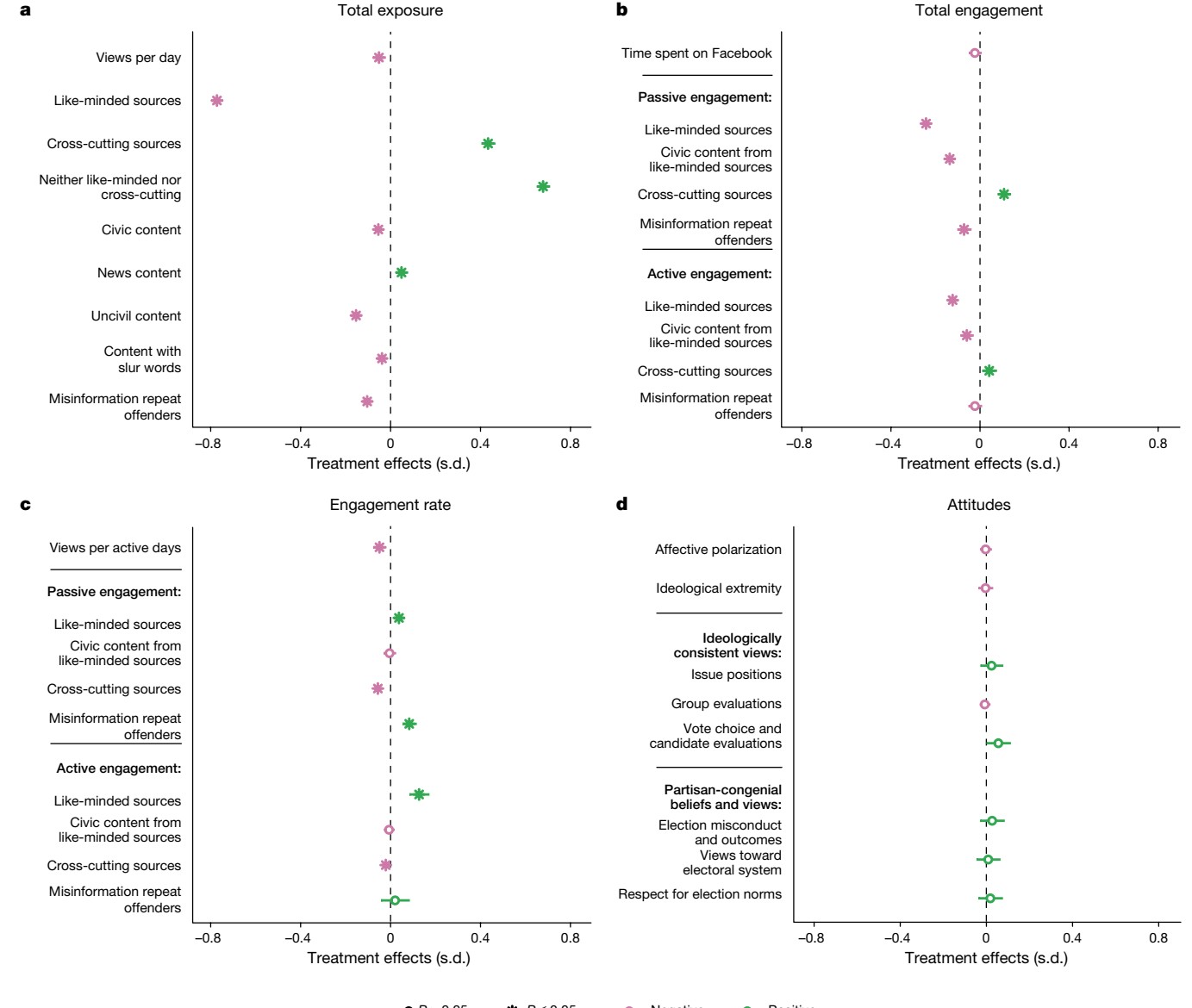

**a** Total exposure

Views per day
Like-minded sources
Cross-cutting sources
Neither like-minded nor cross-cutting
Civic content
News content
Uncivil content
Content with slur words
Misinformation repeat offenders

Treatment effects (s.d.)

**b** Total engagement

Time spent on Facebook
**Passive engagement:**
Like-minded sources
Civic content from like-minded sources
Cross-cutting sources
Misinformation repeat offenders
**Active engagement:**
Like-minded sources
Civic content from like-minded sources
Cross-cutting sources
Misinformation repeat offenders

Treatment effects (s.d.)

**c** Engagement rate

Views per active days
**Passive engagement:**
Like-minded sources
Civic content from like-minded sources
Cross-cutting sources
Misinformation repeat offenders
**Active engagement:**
Like-minded sources
Civic content from like-minded sources
Cross-cutting sources
Misinformation repeat offenders

Treatment effects (s.d.)

**d** Attitudes

Affective polarization
Ideological extremity
**Ideologically consistent views:**
Issue positions
Group evaluations
Vote choice and candidate evaluations
**Partisan-congenial beliefs and views:**
Election misconduct and outcomes
Views toward electoral system
Respect for election norms

Treatment effects (s.d.)

○ P > 0.05    ✴ P ≤ 0.05    ●— Negative    ●— Positive

**Fig. 3 | Effects of reducing Facebook Feed exposure to like-minded sources.**
Average treatment effects of reducing exposure to like-minded sources in the Facebook Feed from 24 September to 23 December 2020. **a**–**c**, Sample average treatment effects (SATE) on Feed exposure and engagement. **b**, Total engagement (for content, the total number of engagement actions). **c**, Engagement rate (the probability of engaging conditional on exposure). **d**, Outcomes of surveys on attitudes, with population average treatment effects (PATEs) estimated using survey weights. Supplementary Information 1.4 provides full descriptions of all outcome variables. Non-bolded outcomes that appear below a bolded header are part of that category. For example, in **d**, 'issue positions', 'group evaluations' and 'vote choice and candidate evaluations' appear below

'ideologically consistent views', indicating that all are measured such that higher values indicate greater ideological consistency. Survey outcome measures are standardized scales averaged across surveys conducted between 4 November and 18 November 2020 and/or 9 December and 23 December 2020. Point estimates are provided in Extended Data Table 3. Sample average treatment effect estimates on attitudes are provided in Extended Data Fig. 4. All effects estimated using ordinary least squares (OLS) with robust standard errors and follow the preregistered analysis plan. Points marked with asterisks indicate findings that are significant ($P < 0.05$ after adjustment); points marked with open circles indicate $P > 0.05$ (all tests are two-sided). $P$ values are false-discovery rate (FDR)-adjusted (Supplementary Information, section 1.5.4).

that content (Fig. 3b), the rate of engagement with content conditional on exposure to it (Fig. 3c), and survey measures of post-election attitudes (Fig. 3d; Extended Data Table 3 reports the corresponding point estimates from Fig. 3; Supplementary Information, section 1.4 provides measurement details).

As seen in Fig. 3a, the reduction in exposure to content from like-minded sources from 53.7% to 36.2% represents a difference of 0.77 s.d. (95% confidence interval: −0.80, −0.75). Total views per day also declined by 0.05 s.d. among treated participants (95% confidence interval: −0.08, −0.02). In substantive terms, the average control group

participant had 267 total content views on a typical day, of which 143 were from like-minded sources. By comparison, 92 out of 255 total content views for an average participant in the treatment condition were from like-minded sources on a typical day (Supplementary Tables 33 and 40).

This reduction in exposure to information from like-minded sources, however, did not lead to a symmetrical increase in exposure to information from cross-cutting sources, which increased from 20.7% in the control group to 27.9% in the treatment group, a change of 0.43 s.d. (95% confidence interval: 0.40, 0.46). Rather, respondents in the treatment

group saw a greater relative increase in exposure to content from sources classified as neither like-minded nor cross-cutting. Exposure to content from these sources increased from 25.6% to 35.9%, a change of 0.68 s.d. (95% confidence interval: 0.65, 0.71).

Figure 3a also indicates that reducing exposure to content from like-minded sources reduced exposure to content classified as containing one or more slur words by 0.04 s.d. (95% confidence interval: −0.06, −0.02), content classified as uncivil by 0.15 s.d. (95% confidence interval: −0.18, −0.13), and content from misinformation repeat offenders (sources identified by Facebook as repeatedly posting misinformation) by 0.10 s.d. (95% confidence interval: −0.13, −0.08). Substantively, the average proportion of exposures decreased from 0.034% to 0.030% for content with slur words (a reduction of 0.01 views per day on average), from 3.15% to 2.81% for uncivil content (a reduction of 1.24 views per day on average), and from 0.76% to 0.55% for content from misinformation repeat offenders (a reduction of 0.62 views per day on average). Finally, the treatment reduced exposure to civic content (−0.05 s.d.; 95% confidence interval: −0.08, −0.03) and increased exposure to news content (0.05 s.d., 95% confidence interval: 0.02, 0.07) (see Supplementary Information, section 1.3 for details on how uncivil content, content with slur words and misinformation repeat offenders are measured).

## Treatment effects on content engagement

We next consider the effects of the treatment (reducing exposure to content from like-minded sources) on how participants engage with content on Facebook. We examine content engagement in two ways, which we call 'total engagement' and 'engagement rate'. Figure 3b presents the effects of the treatment on total engagement with content—the total number of actions taken that we define as 'passive' (clicks, reactions and likes) or 'active' (comments and reshares) forms of engagement. Figure 3c presents effects of the treatment on the engagement rate, which is the probability of engaging with the content that participants did see (that is, engagement conditional on exposure). These two measures do not necessarily move in tandem: as we report below, participants in the treatment group have less total engagement with content from like-minded sources (since they are by design seeing much less of it), but their rate of engagement is higher than that of the control group, indicating that they interacted more frequently with the content from like-minded sources to which they were exposed.

Figure 3b shows that the intervention had no significant effect on the time spent on Facebook (−0.02 s.d., 95% confidence interval: −0.050, 0.004) but did decrease total engagement with content from like-minded sources. This decrease was observed for both passive and active engagement with content from like-minded sources, which decreased by 0.24 s.d. (95% confidence interval: −0.27, −0.22) and 0.12 s.d. (95% confidence interval: −0.15, −0.10), respectively. Conversely, participants in the treatment condition engaged more with cross-cutting sources—passive and active engagement increased by 0.11 s.d. (95% confidence interval: 0.08, 0.14) and 0.04 s.d. (95% confidence interval: 0.01, 0.07), respectively. Finally, we observe decreased passive engagement but no decrease in active engagement with content from misinformation repeat offenders (for passive engagement, −0.07 s.d., 95% confidence interval: −0.10, −0.04; for active engagement, −0.02 s.d., 95% confidence interval: −0.05, 0.01).

When people in the treatment group did see content from like-minded sources in their Feed, however, their rate of engagement was higher than in the control group. Figure 3c shows that, conditional on exposure, passive and active engagement with content from like-minded sources increased by 0.04 s.d. (95% confidence interval: 0.02, 0.06) and 0.13 s.d. (95% confidence interval: 0.08, 0.17), respectively. Furthermore, although treated participants saw more content from cross-cutting sources overall, they were less likely to engage with the content that they did see: passive engagement decreased by 0.06 s.d. (95% confidence interval: −0.07, −0.04) and active engagement decreased by 0.02 s.d.

(95% confidence interval: −0.04, −0.01). The number of content views per days active on the platform also decreased slightly (−0.05 s.d., 95% confidence interval: −0.08, −0.02).

## Treatment effects on attitudes

Finally, we examine the causal effects of reducing exposure to like-minded sources on Facebook on a range of attitudinal outcomes measured in post-election surveys (Fig. 3d). As preregistered, we apply survey weights to estimate PATEs and adjust P values for these outcomes to control the false discovery rate (see Supplementary Information, sections 1.5.4 and 4.7 for details). We observe a consistent pattern of precisely estimated results near zero (open circles in Fig. 3d) for the outcome measures we examine: affective polarization; ideological extremity; ideologically consistent issue positions, group evaluations and vote choice and candidate evaluations; and partisan-congenial beliefs and views about election misconduct and outcomes, views toward the electoral system and respect for election norms (see Supplementary Information, section 1.4 for measurement details). In total, we find that 7 out of the 8 point estimates for our primary outcome measures have values of ±0.03 s.d. or less and are precisely estimated (exploratory equivalence bounds: ±0.1 s.d.; Supplementary Table 60), reflecting high levels of observed power. For instance, the minimum detectable effect in the sample for affective polarization is 0.019 s.d. The eighth result is a less precise null for ideologically consistent vote choice and candidate evaluations (0.056 s.d., equivalence bounds: 0.001, 0.111.)

We also tested the effects of reducing exposure to content from like-minded sources on a variety of attitudinal measures for which we had weaker expectations. Using an exploratory equivalence bounds test, we can again confidently rule out effects of ±0.18 s.d. for these preregistered research questions across 18 outcomes, which are reported in Extended Data Fig. 5 and Supplementary Table 47. An exploratory equivalence bounds analysis also rules out a change in self-reported consumption of media outlets outside of Facebook that we categorized as like-minded of ±0.07 s.d. (Supplementary Tables 59 and 67).

Finally, we examine heterogeneous treatment effects on the attitudes reported in Fig. 3d and the research questions across a number of preregistered characteristics: respondents' political ideology (direction or extremity), political sophistication, digital literacy, pre-treatment exposure to content that is political, and pre-treatment levels of like-minded exposure both as a proportion of respondents' information diet and as the total number of exposures (see Supplementary Information, section 3.9). None of the 272 preregistered subgroup treatment effect estimates for our primary outcomes are statistically significant after adjustment to control the false discovery rate. Similarly, an exploratory analysis finds no evidence of heterogeneous effects by age or number of years since joining Facebook (see Supplementary Information, section 3.9.5).

## Discussion

Many observers share the view that Americans live in online echo chambers that polarize opinions on policy and deepen political divides[6,7]. Some also argue that social media platforms can and should address this problem by reducing exposure to politically like-minded content[38]. However, both these concerns and the proposed remedy are based on largely untested empirical assumptions.

Here we provide systematic descriptive evidence of the extent to which social media users disproportionately consume content from politically congenial sources. We find that only a small proportion of the content that Facebook users see explicitly concerns politics or news and relatively few users have extremely high levels of exposure to like-minded sources. However, a majority of the content that active adult Facebook users in the US see on the platform comes from politically like-minded friends or from Pages or groups with like-minded audiences (mirroring patterns of homophily in real-world networks[15,39]).

This content has the potential to reinforce partisan identity even if it is not explicitly political[14].

Our field experiment also shows that changes to social media algorithms can have marked effects on the content that users see. The intervention substantially reduced exposure to content from like-minded sources, which also had the effect of reducing exposure to content classified as uncivil and content from sources that repeatedly post misinformation. However, the tested changes to social media algorithms cannot fully counteract users' proclivity to seek out and engage with congenial information. Participants in the treatment group were exposed to less content from like-minded sources but were actually more likely to engage with such content when they encountered it.

Finally, we found that reducing exposure to content from like-minded sources on Facebook had no measurable effect on a range of political attitudes, including affective polarization, ideological extremity and opinions on issues; our exploratory equivalence bounds analyses allow us to confidently rule out effects of ±0.12 s.d. We were also unable to reject the null hypothesis in any of our tests for heterogeneous treatment effects across many distinct subgroups of participants.

There are several potential explanations for this pattern of null results. First, congenial political information and partisan news—the types of content that are thought to drive polarization—account for a fraction of what people see on Facebook. Similarly, social media consumption represents a small fraction of most people's information diets[37], which include information from many sources (for example, friends, television and so on). Thus, even large shifts in exposure on Facebook may be small as a share of all the information people consume. Second, persuasion is simply difficult—the effects of information on beliefs and opinion are often small and temporary and may be especially difficult to change during a contentious presidential election[33,40–43]. Finally, we sought to decrease rather than increase exposure to like-minded information for ethical reasons. Although the results suggest that decreasing exposure to information from like-minded sources has minimal effects on attitudes, the effects of such exposure may not be symmetrical. Specifically, decreasing exposure to like-minded sources might not reduce polarization as much as increasing exposure would exacerbate it.

We note several other areas for future research. First, we cannot rule out the many ways in which social media use may have affected participants' beliefs and attitudes prior to the experiment. In particular, our design cannot capture the effects of prior Facebook use or cumulative effects over years; experiments conducted over longer periods and/or among new users are needed (we note, however, that find no evidence of heterogeneous effects by age or years since joining Facebook). Second, although heterogeneous treatment effects are non-existent in our data and rare in persuasion studies in general[44], the sample's characteristics and behaviour deviate in some respects from the Facebook user population. Future research should examine samples that more closely reflect Facebook users and/or oversample subgroups that may be particularly affected by like-minded content. Third, only a minority of Facebook users occupy echo chambers yet the reach of the platform means that the group in question is large in absolute terms. Future research should seek to better understand why some people are exposed to large quantities of like-minded information and the consequences of this exposure. Fourth, our study examines the prevalence of echo chambers using the estimated political leanings of users, Pages, and groups who share content on social networks. We do not directly measure the slant of the content that is shared; doing so would be a valuable contribution for future research. Finally, replications in other countries with different political systems and information environments will be essential to determine how these results generalize.

Ultimately, these findings challenge popular narratives blaming social media echo chambers for the problems of contemporary American democracy. Algorithmic changes that decrease exposure to like-minded sources do not seem to offer a simple solution for those problems. The information that we see on social media may be more a reflection of our identity than a source of the views that we express.

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

# Methods

## Participants

Participants in our field experiment are 73.3% white, 57.3% female, relatively highly educated (50.7% have a college degree), and 54.1% self-identify as Democrats or lean Democrat. They also use Facebook more frequently than the general Facebook population and are exposed to more content from politically like-minded sources (the phenomenon of interest), including civic and news content from like-minded sources, than are other Facebook users (Supplementary Tables 2 and 4–10). Our treatment effect estimates on attitudes therefore apply survey weights created to reflect the population of adult monthly active Facebook users who were eligible for recruitment (see Supplementary Information, section 4.7). The demographic characteristics of the weighted sample are similar to those of self-reported Facebook users in an AmeriSpeak probability sample (Extended Data Table 5).

## Experimental design

Respondents were assigned to treatment or control with equal probability using block randomization (see Supplementary Information, section 4.5 for details; participants were blind to assignment). The Feed of participants in the control condition was not systematically altered. Owing to the difficulty of measuring the political leaning or slant of many different types of content at scale, we instead varied exposure to content based on the estimated political leaning of the source of the information. Using a Facebook classifier, we estimate the political leaning of other users directly (see Supplementary Information, section 1.3 for details). Building on prior research[16,17,23,45,46], we estimate the political leanings of Pages and groups using the political leanings of their audience (group members and Page followers). We classify all users as liberal or conservative using a binary threshold to maximize statistical power, but results are consistent when we exclude respondents with classifications between 0.4 and 0.6 in an exploratory analysis (see Supplementary Information, sections 3.10 and 3.11).

We designed the study to provide statistical power to detect small effects. For instance, our power calculations showed that a final sample size of 24,480 would generate a minimum detectable effect of 1.6 percentage points on vote choice among likely voters (see Supplementary Information, section 4.5).

Randomization was successful: the treatment and control groups do not differ in their demographic characteristics at a rate above what would be expected by chance (see Supplementary Table 5). In total, 82.6% of experimental participants completed at least one post-election survey (23,377 valid completions out of 28,296 eligible participants; see Supplementary Information, section 2.1.3). The final sample consists of respondents who completed at least one post-election survey and did not delete their account or withdraw from the study before data were de-identified. Those who left the study prior to completing a post-election survey do not significantly differ from our final sample (see Supplementary Information, sections 2.1 and 1.2).

## Analyses

All analyses in the main text and in the Supplementary Information follow the preregistration filed at the Open Science Foundation (https://osf.io/3sjy2; see Supplementary Information, section 4.10 except for deviations reported in Supplementary Information, section 4.11). Treatment effect estimates use OLS with robust standard errors and control for covariates selected using the least absolute shrinkage and selection operator[47] (see Supplementary Information, section 1.5.1). As preregistered, our tests of treatment effects on attitudes also apply survey weights to estimate PATEs (see Supplementary Information, section 4.7). Sample average treatment effects, which are very similar, are provided in Supplementary Information, sections 3.2–3.5.

## Ethics

We have complied with all relevant ethical regulations. The overall project was reviewed and approved by the National Opinion Research Center (NORC) Institutional Review Board (IRB). Academic researchers worked with their respective university IRBs to ensure compliance with human subject research regulations in analysing data collected by NORC and Meta and authoring papers based on those findings. The research team also received ethical guidance from Ethical Resolve to inform study designs. More detailed information is provided in Supplementary Information, sections 1.2 and 4.9.

All experimental participants provided informed consent before taking part (see Supplementary Information, section 4.6 for recruitment and consent materials). Participants were given the option to withdraw from the study while the experiment was ongoing as well as to withdraw their data at any time up until their survey responses were disconnected from any identifying information in February 2023. We also implemented a stopping rule, inspired by clinical trials, which stated that we would terminate the intervention before the election if we detected it was generating changes in specific variables related to individual welfare that were much larger than expected. More details are available in Supplementary Information, section 1.2.

None of the academic researchers received financial compensation from Meta for their participation in the project. The analyses were preregistered at the Open Science Foundation (https://osf.io/3sjy2). The lead authors retained final discretion over everything reported in this paper. Meta publicly agreed that there would be no pre-publication approval of papers for publication on the basis of their findings. See Supplementary Information, section 4.8 for more details about the Meta–academic collaboration.

## Reporting summary

Further information on research design is available in the Nature Portfolio Reporting Summary linked to this article.

## Data availability

De-identified data from this project (Meta Platforms, Inc. Facebook Intervention Experiment Participants. Inter-university Consortium for Political and Social Research [distributor], 2023-07-27. https://doi.org/10.3886/9wct-2d24; Meta Platforms, Inc. Exposure to and Engagement with Facebook Posts. Inter-university Consortium for Political and Social Research [distributor], 2023-07-27. https://doi.org/10.3886/9sqy-ny89; Meta Platforms, Inc. Ideological Alignment of Users in Facebook Networks. Inter-university Consortium for Political and Social Research [distributor], 2023-07-27. https://doi.org/10.3886/nvh0-jh41; Meta Platforms, Inc. Facebook User Attributes. Inter-university Consortium for Political and Social Research [distributor], 2023-07-27. https://doi.org/10.3886/vecn-ze56; Stroud, Natalie J., Tucker, Joshua A., NORC at the University of Chicago, and Meta Platforms, Inc. US 2020 FIES NORC Data Files. Inter-university Consortium for Political and Social Research [distributor], 2023-07-27. https://doi.org/10.3886/0d26-d856) are available under controlled access from the Social Media Archive (SOMAR) at the University of Michigan's Inter-university Consortium for Political and Social Research (ICPSR). The data can be accessed via ICPSR's virtual data enclave for university IRB-approved research on elections or to validate the findings of this study. ICPSR will accept and vet all applications for data access. Data access is controlled to protect the privacy of the study participants and to be consistent with the consent form signed by study participants where they were told that their data would be used for "future research on elections, to validate the findings of this study, or if required by law for an IRB inquiry". Requests for data can be made via SOMAR (https://socialmediaarchive.org/); inquiries can be directed to SOMAR staff at somar-help@umich.edu. ICPSR staff will respond to requests for data

within 2–4 weeks of submission. To access the data, the home institution of the academic making the request must complete ICPSR's Restricted Data Agreement. Source data are provided with this paper.

## Code availability

Analysis code from this study (Meta Platforms, Inc. Replication Code for U.S. 2020 Facebook and Instagram Election Study. Inter-university Consortium for Political and Social Research [distributor], 2023-07-27. https://doi.org/10.3886/spb3-g558) is archived at SOMAR, ICPSR (https://socialmediaarchive.org) and made available in the ICPSR virtual data enclave for university IRB-approved research on elections or to validate the findings of this study per the data availability statement above. The data in this study were analysed using R (version 4.1.1), which was executed via R notebooks on JupyterLab (3.2.3). The analysis code imports several R packages available on CRAN, including dplyr (1.0.10), ggplot2 (3.4.0), xtable (1.8-4), aws.s3 (0.3.22), glmnet (4.1.2), SuperLearner (2.0-28), margins (0.3.26) and estimatr (1.0.0).

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

**Acknowledgements** The Facebook Open Research and Transparency (FORT) team provided substantial support in executing the overall project. We are grateful for support on various aspects of project management from C. Nayak, S. Zahedi, I. Rosenn, L. Ahmad, A. Bhalla, C. Chan, A. Gruen, B. Hillenbrand, D. Li, P. McLeod, D. Rice and N. Shah; engineering from Y. Chen, S. Chen, J. Dai, T. Lohman, R. Moodithaya, R. Pyke, Y. Wan and F. Yan; data engineering from B. Xiong, S. Chintha, J. Cronin, D. Desai, Y. Kiraly, T. Li, X. Liu, S. Pellakuru and C. Xie; data science and research from H. Connolly-Sporing, S. Tan and T. Wynter; academic partnerships from R. Mersey, M. Zoorob, L. Harrison, S. Aisiks, Y. Rubinstein and C. Qiao; privacy and legal assessment from K. Benzina, F. Fatigato, J. Hassett, S. Iyengar, P. Mohassel, A. Muzaffar, A. Raghunathan and A. Sun; and content design from C. Bernard, J. Breneman, D. Leto and S. Raj. NORC at the University of Chicago partnered with Meta on this project to conduct the fieldwork with the survey participants and pair the survey data with web tracking data for consented participants in predetermined aggregated forms. We are particularly grateful for the partnership of NORC principal investigator J. M. Dennis and NORC project director M. Montgomery. The costs associated with the research (such as participant fees, recruitment and data collection) were paid by Meta. Ancillary support (for example, research assistants and course buyouts) was sourced by academics from the Democracy Fund, the Guggenheim Foundation, the John S. and James L. Knight Foundation, the Charles Koch Foundation, the Hewlett Foundation, the Alfred P. Sloan Foundation, the Hopewell Fund, the University of Texas at Austin, New York University, Stanford University, the Stanford Institute for Economic Policy Research and the University of Wisconsin-Madison.

**Author contributions** B.N., J.S., E.T., M.W. and P.B. supervised all analyses, analysed data, and wrote the paper. As the academic lead authors, B.N., J.S., E.T. and M.W. had final control rights. P.B. was the lead author at Meta. B.N., J.S., E.T., M.W., D.M. and P.B. designed the study. P.B., D.D., D.F., E.K., Y.M.K., N.M., D.M., B.N., E.T., R.T., C.V.R., A.W. and M.W. contributed study materials (for example, survey questionnaires, classifiers and software). H.A., P.B., A.C.-T., A.F., D.F., M.G., S.G.-B., A.M.G., C.K.d.J., Y.M.K., D.L., N.M., W.M., D.M., B.N., J.P., C.V.R., J.S., N.J.S., E.T., R.T., J.A.T., A.W. and M.W. contributed to the design of the project. P.B., T.B., A.C.-T., A.F., W.M., D.R.T., C.V.R., A.W. and B.X. coordinated the implementation of the experimental intervention and collected and curated all platform data. A.Y.C. and P.B. contributed the figures and tables. E.K. and D.D. contributed to the heterogeneous effects analysis. H.A., M.G., S.G.-B., D.L., N.M., N.J.S. and J.A.T. provided feedback on the manuscript. N.J.S. and J.A.T. were joint principal investigators for the academic involvement on this project, responsible for management and coordination. C.K.d.J., A.F. and W.M. led Meta's involvement on this project and were responsible for management and coordination.

**Competing interests** None of the academic researchers nor their institutions received financial compensation from Meta for their participation in the project. Some authors are or have been employed by Meta: P.B., T.B., A.C.-T., D.D., D.M., D.R.T., C.V.R., A.W., B.X., A.F., C.K.d.J. and W.M. D.D. and C.V.R. are former employees of Meta. All of their work on the study was conducted while they were employed by Meta. The following academic authors have had one or more of the following funding or personal financial relationships with Meta (paid consulting work, received direct grant funding, received an honorarium or fee, served as an outside expert, or own Meta stock): M.G., A.M.G., B.N., J.P., J.S., N.J.S., R.T., J.A.T. and M.W. For additional information about the above disclosures as well as a review of the steps taken to protect the integrity of the research, see Supplementary Information, section 4.8.

**Additional information**
**Correspondence and requests for materials** should be addressed to Brendan Nyhan.

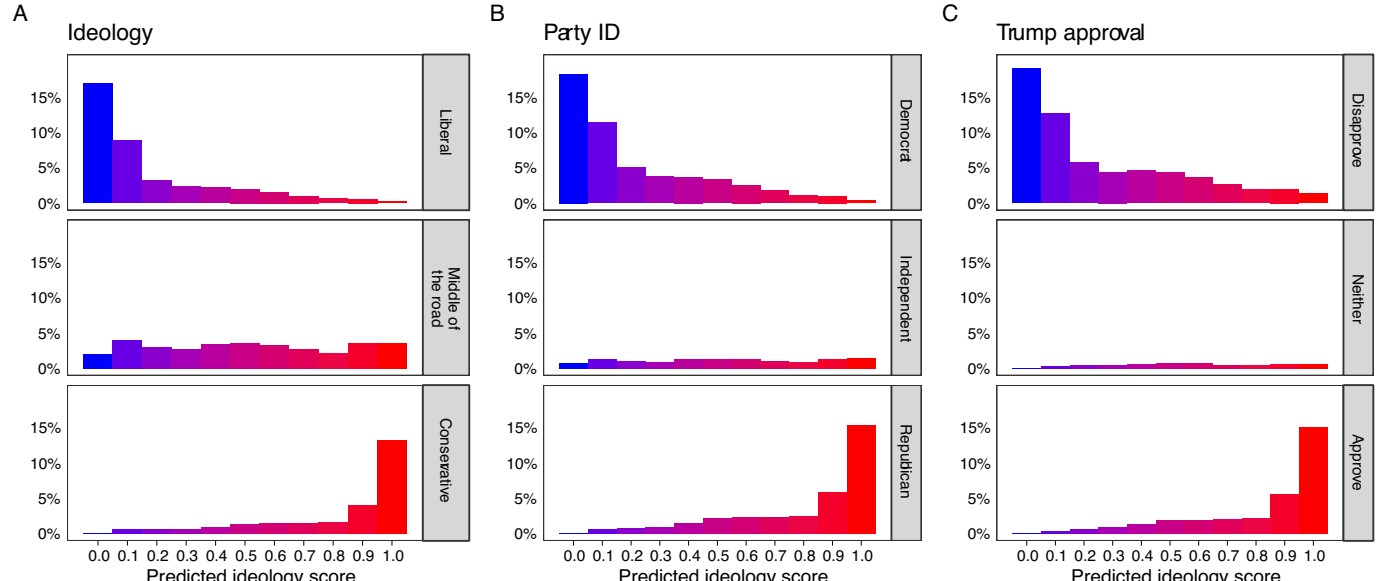

**Extended Data Fig. 1 | Distribution of predicted ideology score by self-reported ideology, party identification, and approval of former president Trump.** Each histograms displays the distribution of respondents' predicted ideology score according to Meta's classifier for Facebook U.S. adult users (see Supplementary Iinformation, section 1.3) by subsets defined by their self-reported political characteristics. The histograms have bins of width equal to 0.10.

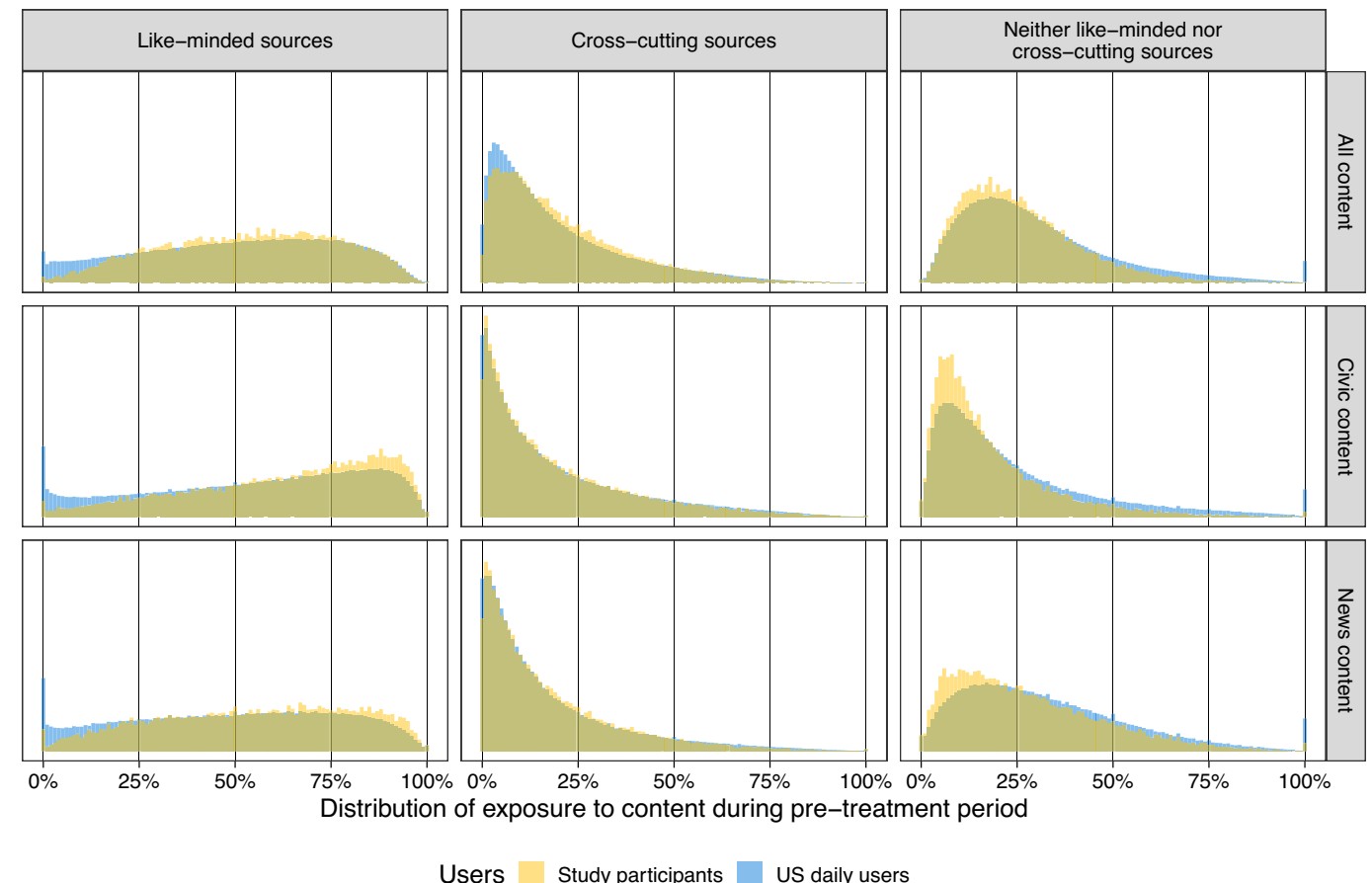

**Extended Data Fig. 2 | Pre-treatment exposure to Facebook Feed content by source type: Study participants and daily Facebook users.** Pre-treatment distribution of Facebook Feed exposure to content from like-minded sources (left column), cross-cutting sources (center column), and those that fall into neither category (right column). Estimates presented for all content (top row) and for content classified as civic (i.e., political; center row) and news (bottom row). Source and content classifications were created using internal Facebook classifiers (see Supplementary Information, section 1.3). The graph includes the distribution of exposure for both study participants and the Facebook population of users age 18+ who logged into Facebook each day in the month prior to August 17, 2020, when the study sampling frame was constructed.

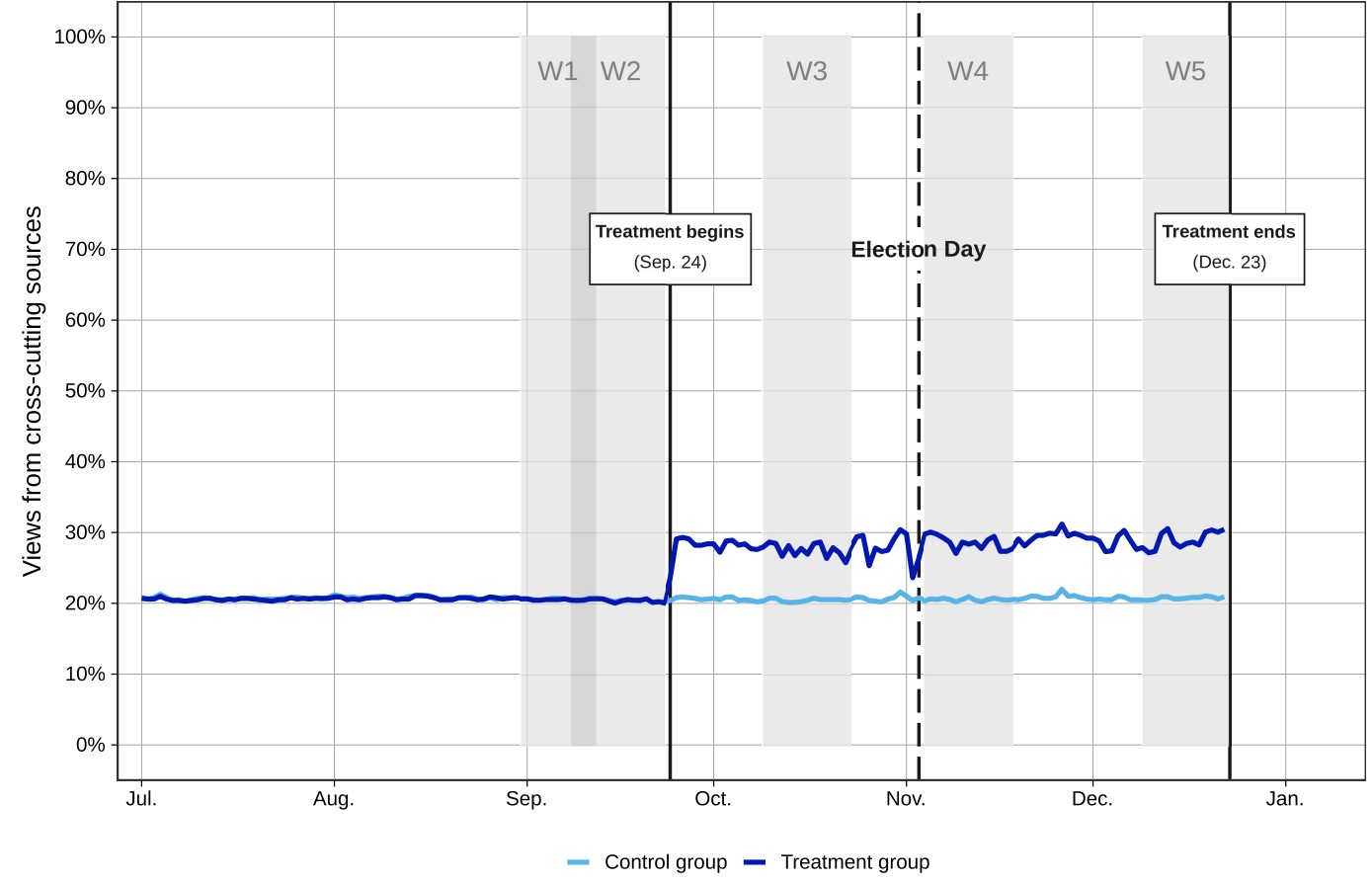

**Extended Data Fig. 3 | Day-level exposure to content from cross-cutting sources in the Facebook Feed by experimental group.** Mean day-level share of respondent views of content from cross-cutting sources by experimental group July 1–December 23, 2020. Sources classified as cross-cutting based on estimates from an internal Facebook classifier at the individual level for users and friends and at the audience level for Pages and groups (see Supplementary Information, section 1.3). W1–W5 indicate survey Waves 1–5; shading indicates wave duration. (Note: Exposure levels briefly decreased on November 2–3 due to a technical problem; see Supplementary Information, section 4.11 for details).

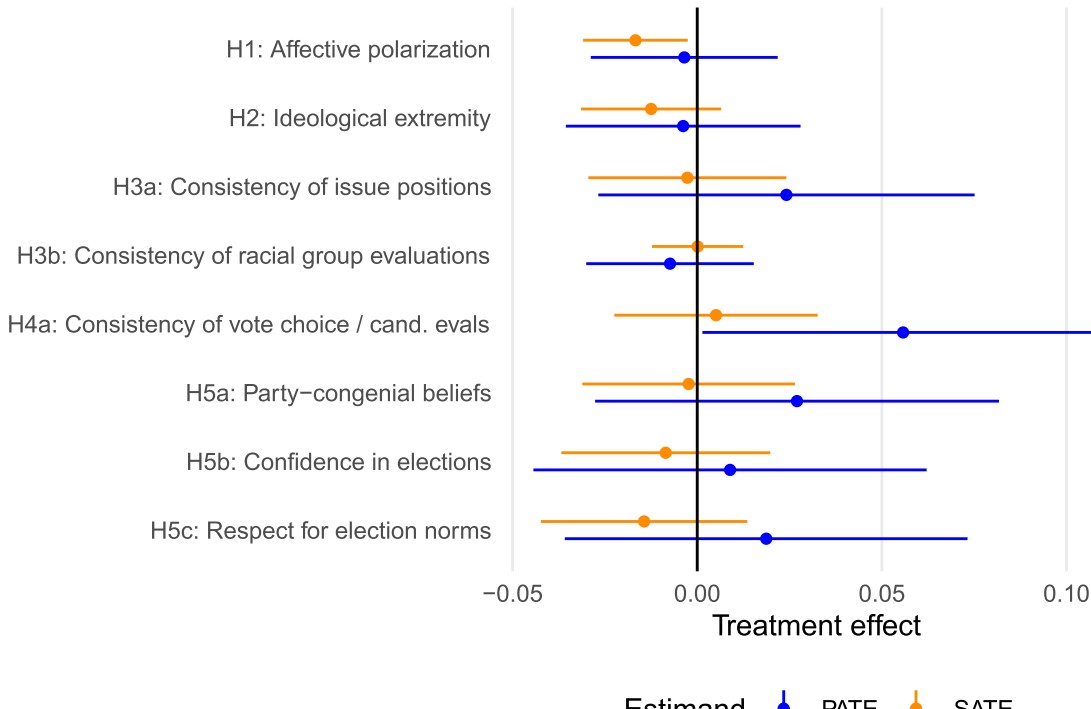

**Extended Data Fig. 4 | Treatment effects on outcomes for primary hypotheses.** Average treatment effects of reducing exposure to like-minded sources in the Facebook Feed from September 24–December 23, 2020. The figure shows OLS estimates of sample average treatment effects (SATE) as well as population average treatment effect (PATE) using survey weights and HC2 robust standard errors. Exposure and engagement outcome measures were measured using Feed behavior by participants. Survey outcome measures are standardized scales averaged across surveys conducted November 4–18, 2020 and/or December 9–23, 2020. Sample size and P values for each estimate are reported in Supplementary Table 47.

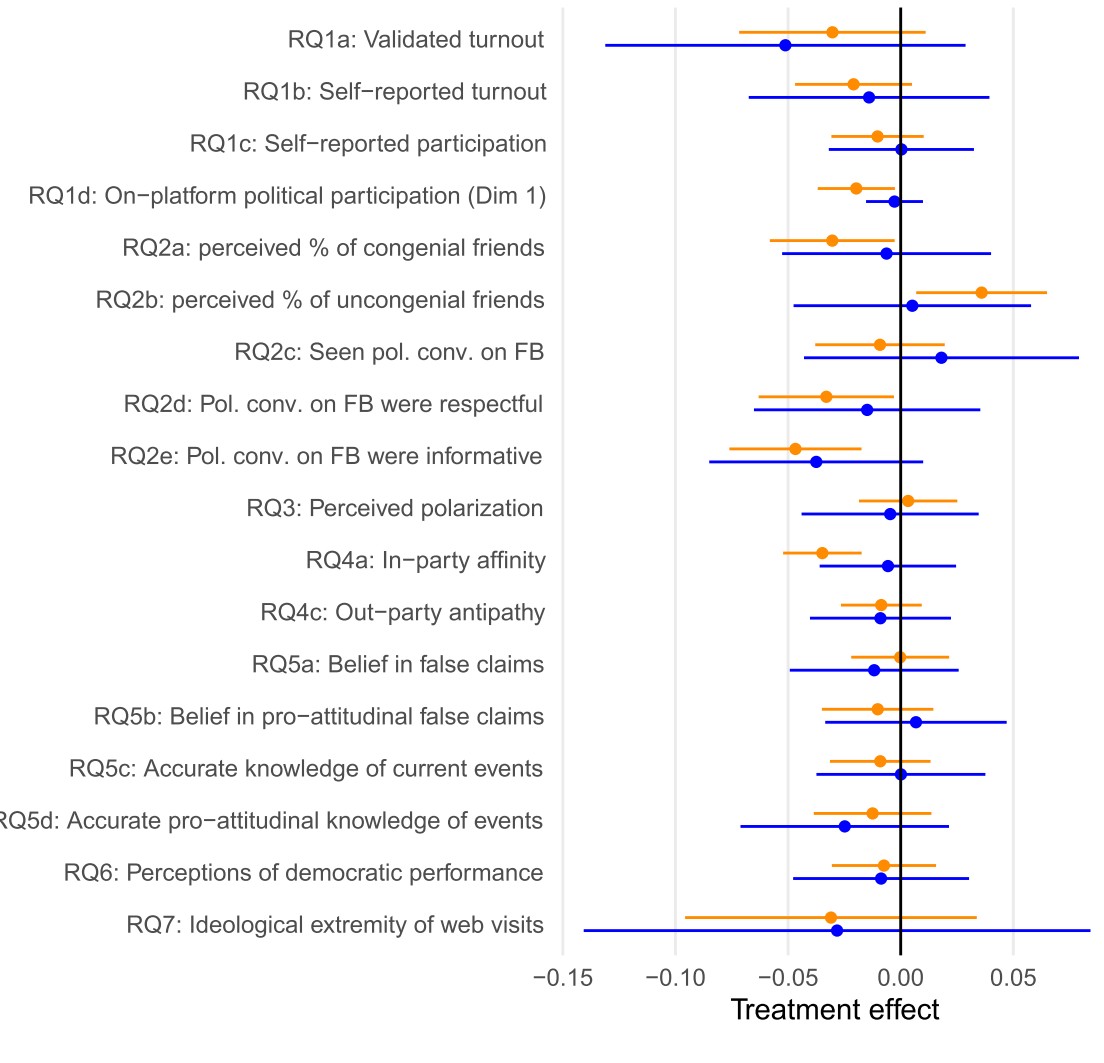

**Extended Data Fig. 5 | Treatment effects on outcomes for research questions.** Average treatment effects of reducing exposure to like-minded sources in the Facebook Feed from September 24–December 23, 2020. The figure shows OLS estimates of sample average treatment effects (SATE) as well as population average treatment effect (PATE) using survey weights and HC2 robust standard errors. Engagement outcome measures were measured using Feed behavior by participants. Survey outcome measures are standardized scales averaged across surveys conducted November 4–18, 2020 and/or December 9–23, 2020, unless indicated otherwise. Sample size and *P* values for each estimate are reported in Supplementary Table 47.

**Extended Data Table 1 | Pre-treatment exposure to Facebook Feed content by source type: Study participants and monthly Facebook users**

| Source type | Respondents | | | | | Monthly active users | | | | |
|---|---|---|---|---|---|---|---|---|---|---|
| | < 25% | 25-50% | 50-75% | > 75% | p50 | < 25% | 25-50% | 50-75% | > 75% | p50 |
| **All content** | | | | | | | | | | |
| Like-minded | 12.6 | 30.2 | 36.8 | 20.4 | 55 | 23.1 | 25.6 | 30.6 | 20.6 | 50 |
| Cross-cutting | 68.1 | 27.4 | 7 | 0.8 | 16.3 | 67.8 | 22 | 8.2 | 2.1 | 15 |
| Other | 55.2 | 31.3 | 7.1 | 0.8 | 23 | 48.7 | 34.4 | 10.7 | 6.2 | 25 |
| **Civic content** | | | | | | | | | | |
| Like-minded | 10 | 33.1 | 50.4 | 39.6 | 67.8 | 23.8 | 20.3 | 26.9 | 29 | 55 |
| Cross-cutting | 71 | 23.4 | 8.9 | 1.8 | 12.7 | 68.3 | 19.4 | 9 | 3.3 | 13 |
| Other | 76.5 | 17.4 | 5.2 | 1 | 13 | 58.6 | 23.1 | 10.2 | 8.1 | 19 |
| **News content** | | | | | | | | | | |
| Like-minded | 15.2 | 28.4 | 38.3 | 26.5 | 56.7 | 27.8 | 24.7 | 27 | 20.6 | 47 |
| Cross-cutting | 75.1 | 20.1 | 6.2 | 1 | 11.7 | 73.1 | 17.3 | 6.9 | 2.7 | 11 |
| Other | 52.4 | 29.8 | 13.7 | 2.1 | 23.8 | 39.7 | 34.7 | 16.3 | 9.3 | 31 |

Pre-treatment exposure by source type among study participants and U.S. adults who logged into Facebook at least once in the month prior to August 17, 2020. The first four columns in each panel of the table report the percentage of users (i.e., respondents or monthly active users) for whom the proportion of content viewed from a given source type (i.e., like-minded, cross-cutting, and sources that fall into neither category) is in the stated range (column). Estimates presented for all content (top set of rows), content classified as civic (i.e., political; center set of rows), and news (bottom set of rows). The final column reports the median (p50), which is approximated to the nearest percentage point among Facebook monthly users for computational efficiency. The denominator for these percentages is all respondents or all monthly active users.

**Extended Data Table 2 | Pre-treatment exposure to Facebook Feed content by source type: Study participants and daily Facebook users**

| Source type | Respondents | | | | | Daily active users | | | | |
|---|---|---|---|---|---|---|---|---|---|---|
| | < 25% | 25-50% | 50-75% | > 75% | p50 | < 25% | 25-50% | 50-75% | > 75% | p50 |
| **All content** | | | | | | | | | | |
| Like-minded | 12.6 | 30.2 | 36.8 | 20.4 | 55 | 19 | 27.2 | 32.7 | 21.1 | 53 |
| Cross-cutting | 68.1 | 27.4 | 7 | 0.8 | 16.3 | 70.4 | 21.8 | 6.9 | 0.9 | 14 |
| Other | 55.2 | 31.3 | 7.1 | 0.8 | 23 | 47.3 | 38.1 | 11.2 | 3.5 | 26 |
| **Civic content** | | | | | | | | | | |
| Like-minded | 10 | 33.1 | 50.4 | 39.6 | 67.8 | 18 | 21 | 29 | 31.9 | 60 |
| Cross-cutting | 71 | 23.4 | 8.9 | 1.8 | 12.7 | 69.2 | 20.1 | 8.4 | 2.3 | 13 |
| Other | 76.5 | 17.4 | 5.2 | 1 | 13 | 63.6 | 23.4 | 8.9 | 4.2 | 18 |
| **News content** | | | | | | | | | | |
| Like-minded | 15.2 | 28.4 | 38.3 | 26.5 | 56.7 | 22.2 | 26.3 | 29.4 | 22.1 | 51 |
| Cross-cutting | 75.1 | 20.1 | 6.2 | 1 | 11.7 | 74 | 17.8 | 6.4 | 1.9 | 11 |
| Other | 52.4 | 29.8 | 13.7 | 2.1 | 23.8 | 42 | 37.7 | 15.7 | 4.7 | 29 |

Pre-treatment exposure by source type among study participants and U.S. adults who logged into Facebook every day in the 30 days preceding August 17, 2020. The first four columns in each panel of the table report the percentage of users (i.e., respondents or daily active users) for whom the proportion of content viewed from a given source type (i.e., like-minded, cross-cutting, and sources that fall into neither category) is in the stated range for (right column). Estimates presented for all content (top set of rows), content classified as civic (i.e., political; center set of rows), and news (bottom set of rows). The final column reports the median (p50), which is approximated to the nearest percentage point among Facebook daily users for computational efficiency. The denominator for these percentages is all respondents or all daily active users.

**Extended Data Table 3 | Treatment effects on total exposure, total engagement, engagement rate, and attitudes**

| Outcome measure | Est. | ATE | 95% CI | SE | N | p | Adjusted p |
|---|---|---|---|---|---|---|---|
| **Panel A: Total exposure** | | | | | | | |
| Like-minded sources | SATE | -0.77 | [-0.797, -0.748] | 0.01 | 23343 | 0.00 | |
| Cross-cutting sources | SATE | 0.43 | [0.405, 0.463] | 0.01 | 23343 | 0.00 | |
| Neither like-minded nor cross-cutting sources | SATE | 0.68 | [0.651, 0.706] | 0.01 | 23343 | 0.00 | |
| Civic content | SATE | -0.05 | [-0.077, -0.031] | 0.01 | 23343 | 0.00 | |
| News content | SATE | 0.05 | [0.023, 0.074] | 0.01 | 23343 | 0.00 | |
| Uncivil content | SATE | -0.15 | [-0.177, -0.130] | 0.01 | 23343 | 0.00 | |
| Content with slur words | SATE | -0.04 | [-0.060, -0.017] | 0.01 | 23343 | 0.00 | |
| Misinformation repeat offenders | SATE | -0.10 | [-0.125, -0.083] | 0.01 | 23343 | 0.00 | |
| Views per day | SATE | -0.05 | [-0.079, -0.025] | 0.01 | 23377 | 0.00 | |
| **Panel B: Total engagement** | | | | | | | |
| Time spent on Facebook | SATE | -0.02 | [-0.050, 0.004] | 0.01 | 23377 | 0.09 | |
| Passive eng. w/like-minded sources | SATE | -0.24 | [-0.265, -0.220] | 0.01 | 23377 | 0.00 | |
| Passive eng. w/civic content from like-minded sources | SATE | -0.14 | [-0.160, -0.113] | 0.01 | 23377 | 0.00 | |
| Passive eng. w/cross-cutting sources | SATE | 0.11 | [0.080, 0.136] | 0.01 | 23377 | 0.00 | |
| Passive eng. w/misinfo. repeat offenders | SATE | -0.07 | [-0.102, -0.042] | 0.02 | 23377 | 0.00 | |
| Active eng. w/like-minded sources | SATE | -0.12 | [-0.146, -0.100] | 0.01 | 23377 | 0.00 | |
| Active eng. w/civic content from like-minded sources | SATE | -0.06 | [-0.087, -0.032] | 0.01 | 23377 | 0.00 | |
| Active eng. w/cross-cutting sources | SATE | 0.04 | [0.010, 0.072] | 0.02 | 23377 | 0.01 | |
| Active eng. w/misinformation repeat offenders | SATE | -0.02 | [-0.051, 0.006] | 0.01 | 23377 | 0.12 | |
| **Panel C: Engagement rate** | | | | | | | |
| Views per active days | SATE | -0.05 | [-0.076, -0.022] | 0.01 | 23355 | 0.00 | |
| Passive eng. w/like-minded sources | SATE | 0.04 | [0.015, 0.060] | 0.01 | 23316 | 0.00 | |
| Passive eng. w/civic content from like-minded sources | SATE | -0.00 | [-0.030, 0.021] | 0.01 | 23148 | 0.73 | |
| Passive eng. w/cross-cutting sources | SATE | -0.06 | [-0.074, -0.039] | 0.01 | 23286 | 0.00 | |
| Passive eng. w/misinformation repeat offenders | SATE | 0.08 | [0.053, 0.113] | 0.02 | 20623 | 0.00 | |
| Active eng. w/like-minded sources | SATE | 0.13 | [0.084, 0.169] | 0.02 | 23316 | 0.00 | |
| Active eng. w/civic content from like-minded sources | SATE | -0.01 | [-0.028, 0.015] | 0.01 | 23148 | 0.53 | |
| Active eng. w/cross-cutting sources | SATE | -0.02 | [-0.037, -0.006] | 0.01 | 23286 | 0.01 | |
| Active eng. w/misinfo. repeat offenders | SATE | 0.02 | [-0.042, 0.082] | 0.03 | 21979 | 0.52 | |
| **Panel D: Attitudes** | | | | | | | |
| Affective polarization | PATE | -0.00 | [-0.029, 0.022] | 0.01 | 22127 | 0.79 | 1.000 |
| Ideological extremity | PATE | -0.00 | [-0.036, 0.028] | 0.02 | 21154 | 0.81 | 1.000 |
| Consistency of issue positions | PATE | 0.02 | [-0.027, 0.075] | 0.03 | 21159 | 0.35 | 1.000 |
| Consistency of group evaluations | PATE | -0.01 | [-0.030, 0.015] | 0.01 | 21162 | 0.52 | 1.000 |
| Consistency of vote choice / candidate evaluations | PATE | 0.06 | [0.001, 0.110] | 0.03 | 21158 | 0.04 | 0.552 |
| Party-congenial beliefs about election | PATE | 0.03 | [-0.028, 0.082] | 0.03 | 20442 | 0.33 | 1.000 |
| Confidence in elections | PATE | 0.01 | [-0.044, 0.062] | 0.03 | 20442 | 0.74 | 1.000 |
| Respect for election norms | PATE | 0.02 | [-0.036, 0.073] | 0.03 | 20390 | 0.50 | 1.000 |

Average treatment effects of reducing exposure to like-minded sources in the Facebook Feed from September 24–December 23, 2020. OLS estimates of sample average treatment effects (SATE) or population average treatment effects using survey weights (PATE) with HC2 robust standard errors. The last two columns report unadjusted and adjusted $P$ values for a two-sided $t$-test for the null hypothesis of no difference between treatment and control groups on each metric. Exposure and engagement outcome measures were measured using Feed behavior by participants. Survey outcome measures are standardized scales averaged across surveys conducted November 4–18, 2020 and/or December 9–23, 2020.

**Extended Data Table 4 | Distribution of exposure by source type (like-minded and cross-cutting) for study participants in the treatment and control groups as well as for monthly active users in both the pre-treatment and treatment periods**

| Metric | Group | p2.5 | p50 | p97.5 | Avg. | SD | N | p(diff!=0) |
|---|---|---|---|---|---|---|---|---|
| % like-minded exposure (pre) | Control | 10.7 | 55.2 | 91.3 | 53.8 | 22.5 | 16046 | – |
| | Treatment | 10.5 | 54.7 | 90.8 | 53.3 | 22.5 | 7180 | p=0.29 |
| | US monthly users | 0 | 50.4 | 93 | 48.4 | 27.2 | 97.8% | – |
| % cross-cutting exposure (pre) | Control | 1.2 | 16.2 | 62.8 | 20.6 | 16.7 | 16046 | – |
| | Treatment | 1.2 | 16.5 | 61.8 | 20.8 | 16.5 | 7180 | p=0.62 |
| | US monthly users | 0 | 14.7 | 72.4 | 20.8 | 19.7 | 97.8% | – |
| % like-minded exposure | Control | 11 | 54.9 | 91.3 | 53.7 | 22.3 | 16046 | – |
| | Treatment | 5.6 | 32.8 | 81.4 | 36.2 | 20.8 | 7180 | p=0.00 |
| | US monthly users | 0 | 49.6 | 93.4 | 48.6 | 26.6 | 96.9% | – |
| % cross-cutting exposure | Control | 1.2 | 16.3 | 63.3 | 20.7 | 16.7 | 16046 | – |
| | Treatment | 2.1 | 24.6 | 70.3 | 27.9 | 18.6 | 7180 | p=0.00 |
| | US monthly users | 0 | 15.2 | 68.6 | 20.5 | 18.6 | 96.9% | – |

151 observations (0.65%) dropped by listwise deletion. US monthly users indicates the set of US adults who logged onto Facebook at least once in the 30 days preceding 17 August 2020. The last column reports the unadjusted $P$ value from a two-sided test of the hypothesis of no difference between treatment and control groups on each metric, computed using the baseline OLS model (see Section 1.5.1).

**Extended Data Table 5 | Comparison of study participants, active Facebook users, and U.S. population**

| Demographic | Category | Participants | | Amerispeak | |
|---|---|---|---|---|---|
| | | Unweighted | Weighted | FB users | All |
| Age | 18–29 | 18.1 | 21.9 | 21.4 | 20.5 |
| | 30–44 | 42.6 | 38.7 | 28.5 | 25.6 |
| | 45–65 | 32.5 | 32.0 | 34.1 | 33.9 |
| | >65 | 6.8 | 7.4 | 16.0 | 19.9 |
| Race | Asian or Pacific Islander | 2.2 | 3.0 | 4.3 | 4.5 |
| | Black, Non-Hispanic | 6.7 | 8.5 | 11.6 | 11.9 |
| | Hispanic | 12.0 | 15.7 | 16.5 | 16.7 |
| | Other | 5.7 | 7.8 | 4.1 | 4.1 |
| | White | 73.3 | 65.0 | 63.5 | 62.8 |
| Gender | Female | 57.3 | 49.7 | 55.5 | 51.7 |
| | Male | 41.9 | 49.4 | 44.5 | 48.3 |
| | Other | 0.8 | 0.9 | 0.0 | 0.0 |
| Party ID | Strong Democrat | 25.1 | 20.9 | 20.9 | 21.2 |
| | Not very strong Democrat | 13.7 | 13.6 | 16.6 | 16.3 |
| | Independent, but closer to Democrat | 15.2 | 13.6 | 10.9 | 11.3 |
| | Independent | 12.4 | 11.4 | 11.4 | 11.5 |
| | Independent, but closer to Republican | 10.0 | 12.3 | 9.6 | 9.6 |
| | Not very strong Republican | 10.4 | 13.0 | 13.6 | 13.0 |
| | Strong Republican | 13.1 | 15.2 | 17.0 | 17.1 |
| Ideology | Very liberal | 16.6 | 12.8 | 9.6 | 9.9 |
| | Somewhat liberal | 24.6 | 21.0 | 19.6 | 19.6 |
| | Middle of the road | 33.4 | 35.8 | 34.9 | 33.8 |
| | Somewhat conservative | 17.6 | 21.2 | 23.8 | 24.2 |
| | Very conservative | 7.7 | 9.1 | 12.0 | 12.6 |
| Income tercile | $1^{st}$ | 33.1 | 38.5 | 46.2 | 46.0 |
| | $2^{nd}$ | 34.4 | 32.2 | 33.2 | 32.7 |
| | $3^{rd}$ | 32.6 | 29.2 | 20.7 | 21.4 |
| Education | Does not have college degree | 49.3 | 64.4 | 65.5 | 65.7 |
| | Has college degree | 50.7 | 35.6 | 34.5 | 34.3 |
| N | | 23377 | 23377 | 8837 | 12001 |

Demographics and political attitudes of study participants without survey weights applied (first column), study participants with survey weights created to reflect the population of adult monthly active Facebook users who were eligible for recruitment (second column), respondents who report using Facebook in an AmeriSpeak probability sample with survey weights applied (third column), and all respondents in the AmeriSpeak probability sample with survey weights applied (fourth column). See SI Section 4.5 for details on the sample.

# Reporting Summary

## Statistics

For all statistical analyses, confirm that the following items are present in the figure legend, table legend, main text, or Methods section.

| n/a | Confirmed | |
|---|---|---|
| ☐ | ☒ | The exact sample size (*n*) for each experimental group/condition, given as a discrete number and unit of measurement |
| ☐ | ☒ | A statement on whether measurements were taken from distinct samples or whether the same sample was measured repeatedly |
| ☐ | ☒ | The statistical test(s) used AND whether they are one- or two-sided<br>*Only common tests should be described solely by name; describe more complex techniques in the Methods section.* |
| ☐ | ☒ | A description of all covariates tested |
| ☐ | ☒ | A description of any assumptions or corrections, such as tests of normality and adjustment for multiple comparisons |
| ☐ | ☒ | A full description of the statistical parameters including central tendency (e.g. means) or other basic estimates (e.g. regression coefficient) AND variation (e.g. standard deviation) or associated estimates of uncertainty (e.g. confidence intervals) |
| ☐ | ☒ | For null hypothesis testing, the test statistic (e.g. *F*, *t*, *r*) with confidence intervals, effect sizes, degrees of freedom and *P* value noted<br>*Give P values as exact values whenever suitable.* |
| ☒ | ☐ | For Bayesian analysis, information on the choice of priors and Markov chain Monte Carlo settings |
| ☒ | ☐ | For hierarchical and complex designs, identification of the appropriate level for tests and full reporting of outcomes |
| ☐ | ☒ | Estimates of effect sizes (e.g. Cohen's *d*, Pearson's *r*), indicating how they were calculated |

*Our web collection on statistics for biologists contains articles on many of the points above.*

## Software and code

Policy information about availability of computer code

| | |
|---|---|
| Data collection | Data collection was carried out by Meta and NORC, an independent survey research organization at the University of Chicago. Meta recruited most participants and collected on-platform data. NORC carried out all surveys associated with the project, recruited additional survey panelists, collected all supplemental data outside of the Facebook/Instagram on-platform data, and removed any direct identifiers before linking to the survey data and sharing with the research team.<br><br>On-platform behavioral data were collected via Meta's internal systems for logging user behavior. Survey data were collected by NORC using their existing survey infrastructure. To collect the passive measurement data, NORC partnered with two vendors: MDI Global and RealityMine. Users who consented to passive data tracking were asked to install an app and use a virtual private network (VPN) on their mobile or desktop devices to collect data about the number of visits and time spent on different web domains as well as usage and time spent on apps on their mobile device. The app was developed by MDI Global and the VPN was developed and maintained by RealityMine. Both firms collected the passive tracking data and sanitized, truncated, and/or categorized the URLs to minimize the risk of sharing any additional personally identifiable information (PII). |
| Data analysis | Analysis code from this study is archived in the Social Media Archive (SOMAR) at ICPSR (https://socialmediaarchive.org) and made available in the ICPSR virtual data enclave for university IRB-approved research on elections or to validate the findings of this study per the data availability statement above. The data in this study was analyzed using R (version 4.1.1), which was executed via R notebooks on JupyterLab (3.2.3). The analysis code imports several R packages available on CRAN, including dplyr (1.0.10), ggplot2 (3.4.0), xtable (1.8-4), aws.s3 (0.3.22), glmnet (4.1.2), SuperLearner (2.0-28), margins (0.3.26), and estimatr (1.0.0). |

For manuscripts utilizing custom algorithms or software that are central to the research but not yet described in published literature, software must be made available to editors and reviewers. We strongly encourage code deposition in a community repository (e.g. GitHub). See the Nature Portfolio guidelines for submitting code & software for further information.

# Data

Policy information about availability of data

All manuscripts must include a data availability statement. This statement should provide the following information, where applicable:

- Accession codes, unique identifiers, or web links for publicly available datasets
- A description of any restrictions on data availability
- For clinical datasets or third party data, please ensure that the statement adheres to our policy

De-identified data from this project (Meta Platforms, Inc. Facebook Intervention Experiment Participants. Inter-university Consortium for Political and Social Research [distributor], 2023-07-27. https:// doi.org/10.3886/9wct-2d24; Meta Platforms, Inc. Exposure to and Engagement with Facebook Posts. Inter-university Consortium for Political and Social Research [distributor], 2023-07-27. https://doi. org/10.3886/9sqy-ny89; Meta Platforms, Inc. Ideological Alignment of Users in Facebook Networks. Inter-university Consortium for Political and Social Research [distributor], 2023-07-27. https://doi. org/10.3886/nvh0-jh41; Meta Platforms, Inc. Facebook User Attributes. Inter-university Consortium for Political and Social Research [distribu- tor], 2023-07-27. https://doi.org/10.3886/vecn-ze56; Stroud, Natalie J., Tucker, Joshua A., NORC at the University of Chicago, and Meta Plat- forms, Inc. US 2020 FIES NORC Data Files. Inter-university Consortium for Political and Social Research [distributor], 2023-07-27. https://doi. org/10.3886/0d26-d856) is available under controlled access from the Social Media Archive (SOMAR) at the University of Michigan's Inter-university Consortium for Political and Social Research (ICPSR). The data can be accessed via ICPSR's virtual data enclave for university IRB-approved research on elections or to validate the findings of this study. ICPSR will accept and vet all applications for data access. Data access is controlled to protect the privacy of the study participants and to be consistent with the consent form signed by study participants where they were told that their data would be used for "future research on elections, to validate the findings of this study, or if required by law for an IRB inquiry." Requests for data can be made via the SOMAR website (https://socialmediaarchive.org); inquiries can be directed to SOMAR staff at somar-help@umich.edu. ICPSR staff will respond to requests for data within 2-4 weeks of submission. To access the data, the home institution of the academic making the request must complete ICPSR's Restricted Data Agreement.

The categorization methods described in section S6.2 rely on two open-source labeled datasets: a list of slurs sourced from Hatebase (hatebase.org) and the Racial Slur Database (rsdb.org) that was compiled by Siegel et al. (2021) and two sets of social media posts annotated by human coders by whether they are perceived as uncivil or not (Theocharis et al. 2020; Davidson et al. 2020). We describe these in more detail in Section 4.3.2 of the SI, "Other classifiers and categorization methods."

# Human research participants

Policy information about studies involving human research participants and Sex and Gender in Research.

| | |
|---|---|
| Reporting on sex and gender | We confirm that we do not use the terms gender or sex in the main text. Several analyses in the SI employ gender, which is measured via survey self-report (male, female, and other). Gender was determined based on survey self-reports. Informed consent was provided prior to collecting survey data.<br><br>For our main findings, gender is included as a candidate covariate (included covariates were selected via lasso). In addition, we conduct several analyses that examine how the effects of the treatment on exposure to different types of content vary by gender (as well as a variety of other subgroups). These analyses are available in Section S3.9.5 of the SI, "HTE analysis for exploratory moderators." |
| Population characteristics | Our experimental sample is 73.3% white (2.2% Asian or Pacific Islander, 6.7% Black, Non-Hispanic, 12% Hispanic, 5.7% Other), 57.3% female (41.9% male and 0.8% Other), and relatively highly educated (50.7% have a college degree. With regard to self-reported party identification, the sample is Democratic-leaning (54.1% self-identify as Democrats or lean Democrat, 33.5% self-identify as Republicans or lean Republican, and 12.4% are Independents leaning toward neither party). With regard to self-reported ideology, 41.2% were very or somewhat liberal, 33.4% "middle of the road" and 25.3% were somewhat or very conservative. |
| Recruitment | We summarize the recruitment strategy below (it is briefly described in the main text as well). Further details are provided in Section S4.6 in the SI. At the top of their Facebook feed, randomly selected participants saw a recruitment message asking them if they would like to share their opinion. Those clicking "Start Survey" were directed to a consent form. Participants gave their consent to participate using an IRB-approved consent form that outlined the study procedure, benefits and risks, and compensation.<br><br>Our analyses show that participants use Facebook more frequently than the general Facebook population and are exposed to more politically like-minded content (the phenomenon of interest), including like-minded civic and news content, than are other Facebook users. To address potential self-selection bias, our treatment effect estimates on attitudes apply survey weights created to reflect the population of adult monthly active Facebook users who were eligible for recruitment.<br><br>We also note that randomization into the treatment and control condition was successful, with no statistically significant differences between the groups on 25 out of 26 characteristics (see Table S5 in the SI). We also provide evidence there showing no indication of differential attrition across waves by treatment status and quartile of pre-treatment exposure to content from like-minded sources (see Table S6). As such, attrition bias should not impact the results. |
| Ethics oversight | We have complied with all relevant ethical regulations. The overall project was reviewed and approved by the NORC IRB. Academic researchers worked with their respective university IRBs to ensure compliance with human subjects research regulations in analyzing data collected by NORC and Meta and authoring papers based on those findings. The research team also received ethical guidance from the independent firm Ethical Resolve to inform study designs. More detailed information is provided in Sections S1.2 and S4.9 of the SI. |

All participants provided informed consent before taking part (see SI Section S4.6 for recruitment and consent materials). Participants were given the option to withdraw from the study while the experiment was ongoing as well as to withdraw their data at any time up until their survey responses were disconnected from any identifying information in February 2023. We also implemented a stopping rule, inspired by clinical trials, which stated that we would terminate the intervention before the election if we detected it was generating changes in specific variables related to individual welfare that were much larger than expected. More details are available in SI Section S1.2.

None of the academic researchers received compensation from Meta for their participation in the project. The analyses were preregistered at the Open Science Foundation. The lead authors retained final discretion over everything reported in this paper. Meta publicly agreed that there would be no pre-publication approval of papers for publication on the basis of their findings. See SI Section S4.8 for more details about the Meta-academic collaboration.

Note that full information on the approval of the study protocol must also be provided in the manuscript.

# Field-specific reporting

Please select the one below that is the best fit for your research. If you are not sure, read the appropriate sections before making your selection.

☐ Life sciences   ☒ Behavioural & social sciences   ☐ Ecological, evolutionary & environmental sciences

For a reference copy of the document with all sections, see nature.com/documents/nr-reporting-summary-flat.pdf

# Behavioural & social sciences study design

All studies must disclose on these points even when the disclosure is negative.

| Study description | We rely on an over-time experimental design. Respondents were assigned to treatment or control with equal probability using block randomization. The News Feed of participants in the control condition was not systematically altered. For participants assigned to treatment, we downranked content from friends, Groups, and Pages who were predicted to share the participant's estimated political leaning. The details on the design are presented in the Design section of the main text, as well as in Section S1 in the SI, "Materials and Methods." The data are quantitative. |
|---|---|
| Research sample | Participants in the experiment consisted of U.S. Facebook users age 18 and over who agreed to participate in a study of social media and politics and completed both baseline survey waves. They were recruited via survey invitations placed at the top of their feeds and remunerated for their participation (details on sampling are provided in Section S8 of the Supplementary Information). The sampling frames included all Facebook monthly active U.S.-based users 18 years of age or older eligible to receive general surveys on a given platform (these represent a random set of users from the overall Facebook populations) as of August 17, 2020. Participants were asked to confirm they were over 18 years of age and lived in the United States as part of the recruitment process. Platform-wide statistics were provided as aggregate data for U.S.-based users 18 years of age or older who were active at least once per month, a standard social media measure often known as monthly active users/people (MAP/MAU). This sample represents the subset of adults (18+ years old) in the 231 million people who accessed Facebook every month during this period. |
| Sampling strategy | Below we summarize our sampling strategy. Further details are available in Section S8 of the SI, "Sampling, strata definitions, randomization, and power analyses."<br><br>The sampling approach was designed to achieve specific sample targets across different stages of the study. The sample targets were chosen to achieve desired minimum detectable effect sizes (MDEs) across different subgroups. The sampling frames included all Facebook monthly active U.S.-based users 18 years of age or older eligible to receive general surveys on a given platform (these represent a random set of users from the overall Facebook populations) as of August 17, 2020. The sample stratification took into account the following variables: number of days a user was active on a given platform, a user's predicted census region, whether the user is predicted to live in a battleground state, a user's predicted ideology, and the census ethnic/racial composition in the zip code in which a user is predicted to live. Sampling probabilities were computed to achieve specific sample distributions for the set of demographics encoded in the stratification step across each of the samples of interest. The sampling probabilities took into account (a) differential nonresponse across different demographics (see section S10) and (b) the desired sample size across the different studies. The frame was adjusted as we reviewed the incoming data (see section S8.3). We designed our sampling approach with the goal of recruiting the minimum number of respondents required to detect meaningful effect sizes (see SI Section S8.4, "Power calculations," for more detail). |
| Data collection | Data was collected from participants on their own devices (e.g., mobile phones and computers). The survey vendor we used (NORC) was blind to the experimental condition of each subject as well as our hypotheses. |
| Timing | Data collection started on August 31, 2020. Two surveys were fielded pre-treatment: Wave 1 (August 31-September 12) and Wave 2 (September 8-23). The treatment ran from September 24-December 23. During the treatment period, three more surveys were administered: Wave 3 (October 9-23), Wave 4 (November 4-18), and Wave 5 (December 9-23). This process is outlined in the "Field experiment among consenting U.S. Facebook users section of the main manuscript." |

| | |
|---|---|
| Data exclusions | Data from 25 users (0.1% of the sample) for whom no classifier prediction for ideology was available were excluded from the analyses because it would not be possible to determine whether certain sources of online content were congenial or cross-cutting for those participants. Details on this exclusion are available in SI Section S1.3, "Classifiers." This exclusion criteria was preregistered. |
| Non-participation | We detail information about recruitment and response rates for the collaboration in the Supplementary Information S9.4. In total, 75,318 participants were randomized into one of the experimental conditions within the collaboration. Of these, 8 (0.01%) withdrew from the study after completing a post-treatment wave, and 1,369 (1.8%) deleted or deactivated their Facebook account since the study was completed. Data from these participants are not included in the analyses in this paper. This information is detailed in S2.1.3 of the SI, "Deleted Accounts and Study Withdrawals." |
| Randomization | Respondents were randomly assigned to treatment or control with probabilities that maximized statistical power using block randomization. A combination of survey-based pre-treatment outcomes and Facebook data were used to define the blocks in the sample of interest. The full details are available in Section S9.3 of the SI, "Randomization." |

# Reporting for specific materials, systems and methods

We require information from authors about some types of materials, experimental systems and methods used in many studies. Here, indicate whether each material, system or method listed is relevant to your study. If you are not sure if a list item applies to your research, read the appropriate section before selecting a response.

## Materials & experimental systems

| n/a | Involved in the study |
|---|---|
| ☒ | Antibodies |
| ☒ | Eukaryotic cell lines |
| ☒ | Palaeontology and archaeology |
| ☒ | Animals and other organisms |
| ☒ | Clinical data |
| ☒ | Dual use research of concern |

## Methods

| n/a | Involved in the study |
|---|---|
| ☒ | ChIP-seq |
| ☒ | Flow cytometry |
| ☒ | MRI-based neuroimaging |

