## [Peer Review File · Nature]

Manuscript Title: Like-minded sources on Facebook are prevalent but not polarizing

Reviewer Comments & Author Rebuttals

Reviewer Reports on the Initial Version:

Referees' comments:

Referee #1 (Remarks to the Author):

This is an important and careful study that deserves to be published. I have a few suggestions:

-Is it possible to provide more information about the intervention in the main text - by what magnitude did the intervention *attempt* to decrease exposure to like-minded content?

-The abstract says: "The intervention increases exposure to content from cross-cutting sources and decreases exposure to uncivil language". This language, and similar language in the main body of the paper, is not clear on whether this is something the intervention was supposed to accomplish mechanistically, or if this is a substantive finding.

-The paper said all users were assigned political leaning scores. How is it possible for a user to have a political leaning score if the user never engaged with or posted political content?

-“Despite the prevalence of like-minded sources in what people see on Facebook, extreme “echo chamber” patterns of exposure are relatively rare. Just 20.6% of Facebook users get over 75% of their exposures from like-minded sources.” Is 20.6% rare?

-Figure 3 – It is hard to interpret an effect on certain variables, eg “vote choice and candidate evaluation” – the reader needs to know what the variable is, and what higher v lower means. eg For figure C is “active engagement” the rate or total? I recommend adding full variable definitions to the caption.

-A number of concepts are used in the paper that are not defined in the main body of the paper. I imagine the authors are cognizant of space constraints, but I think it’s important that these definitions are pushed into the main text. One example: “cross-cutting sources”. Another example is that the introduction says: “First, the majority of the content that active adult Facebook users in the U.S. see comes from like-minded friends, Pages, and Groups, although only small fractions of this content are categorized as news or explicitly about politics.” I believe like-minded is defined vis a vis politics later in the paper, but I think it would be clearer if it was defined on first use. I don’t believe like-minded is ever defined for non-political issues/topics.

Referee #2 (Remarks to the Author):

This article presents a field experiment on Facebook that tested if reducing the exposure to sources of similar political alignment of users had an effect that reduced indicators related to political polarization. It combines large-scale data analysis and controlled experimentation to illustrate the problem of exposure to like-minded sources and how a three-month change in the

Facebook News feed algorithm does not produce noticeable results in the main outcomes related to polarization.

The design of the experiment, based on a three-month treatment and survey waves in a four-month period, is impressive and reaches an extremely large sample. The analysis methodology is carefully designed, with a pre-registration and a collaboration plan that clears many doubts that scientists had about this kind of industry-academia collaboration. I was one of the skeptical ones and I never thought I would see a collaboration this ambitious and well-executed between academics and a company with a clear stake in the research question. This might be the most ambitious social media field experiment I have seen and I think that this article has the potential to be one of the most influential papers about social media of this decade.

The article frames its motivation around the public discussion regarding the role of social media in political polarization, in particular, the potential role of "echo chambers". While the authors do not precisely define what an echo chamber is, they refer to both popular and academic sources that describe it as a phenomenon that makes social media users receive ideologically-confirming information over other views. This definition does not distinguish social drivers (e.g. political homophily in online social networks) from algorithmic mechanisms (e.g. the Facebook News feed algorithm or search ranking algorithms) and this distinction is one of the core features of this study that makes it especially important. This experiment is unlike others that manipulated only the social component, for example (Bail et al, 2018), where the treatment asked participants to follow an account that posted cross-cutting content. It is also different from experiments asking people to stop using Facebook, which could not disentangle the social and algorithmic parts of the echo chamber. Especially when comparing with offline interaction, this differentiation between social drivers and algorithmic mechanisms is especially important as it is a possible explanation for existing conflicting results in the literature. Here, I refer to the results of (Allcott et al, 2020) versus those of (Asimovic et al, 2021), which could be explained by Bosnia and Herzegovina having stronger offline (ethnic) echo chambers than the US. I think the treatment in the experiment of this paper is much more interesting than just "opening the echo chamber", to my knowledge is the first experiment that has manipulated the algorithm directly and nothing else, thus being an unprecedented example that can isolate the role of algorithm mechanisms from any social driver that is also changed when, for example, people stop using Facebook.

Among some suggestions for improvement of the article, I think the extent of the effect of the treatment in the exposure to like-minded sources is hard to assess in the current manuscript. It takes until the end of page 3 to see that the reduction of exposure to like-minded sources was for about one-third, this could be mentioned more clearly in the abstract and at times when describing the intervention. The reader needs to be aware that the downranking of like-minded sources was not complete and still users saw more like-minded content than cross-cutting content (36.2% vs 27.9%) and whether this differs from what we would expect from a plain chronological ranking based on friends. I suspect that this "no algorithm" case would still be very skewed toward like-minded sources and the actual effect of the algorithm change is stronger than having no algorithm but still less than showing a random sample from all of US Facebook.

A change that would help in this respect is to show in Figure 2 the mean day-level share of respondent views of content from cross-cutting sources, perhaps with another color, so that the reader can see how that exposure has changed too and to which extent it stays below the exposure to like-minded sources even after the beginning of the treatment.

The observational study of the full sample of Facebook users is impressive and something that seizes the best opportunities of a collaboration like this one. However, the size of this full sample is never disclosed in the article. Even table S3, which directly compares the survey respondents and the US MAP sample, reports the exact N of each calculation on the survey sample but the N for US MAP is always reported as "100%". To assess the results of the full sample, we need to know the number of users included in it (at least approximately) and more descriptive statistics, especially about ideology scores. To interpret the precision and recall values of the ideology classifier, we

need to know the sample size of users labeled as each, or even better: the histogram of ideology scores. This will also allow a reader to see whether the 0.5 threshold could be biased or how sensitive the distribution is to the 0.4/0.6 thresholds used later. A report on the full sample size would also help to understand the fraction of the US population that is included in this observational study. The article makes a great motivation point when citing that Twitter is only used by 23% of the public, but with the data reported here, we should have a direct number of the percentage of the above-18 public in the US that is included in the study.

I acknowledge that this kind of request might be hard to fulfill in an industry-academia collaboration and the last thing I want is to delay the article with unreasonable requests but I think that sample size is an important metric to report for research transparency, also for observational digital-trace data analysis. I think that the Meta/academic agreement would allow this kind of reporting of aggregate numbers that bring no risk to privacy. This might be spelled out more in detail in the linked OSF repository (<https://osf.io/7wpgd>) but it was not online at the time of my review and I have the impression that this would not require any review by Meta since reporting these numbers are not an issue regarding confidential or personally identifiable information or to play a role in abiding by existing legal obligations.

On a related comment, the paper is rather clear about Meta not having pre-publication approval and only being entitled to review the paper regarding these sensitive privacy and legal reasons, but what remains unclear in the article is if Meta did in fact review the article beyond the authors listed in the study. The separation between changes requested for research or for legal/privacy obligations is blurry and, while it's clear that the academic authors here did their best and the process was a great example of transparency, a reader needs to know more of the extent to which Meta did take a role in reviewing the content of the paper and not that it was just a possibility. The reporting of sample sizes in the observational study is also important to contextualize the finding that just 20.6% of Facebook users get over 75% of their exposures from like-minded sources, which is discussed in the article as evidence that is not consistent with the worst fears about echo chambers. While this is not as bad as what is known from offline interaction nor as prevalent as how the echo chamber has been portrayed, the picture is different if we think about Facebook as news media. Assuming that Facebook had about 250 Million MAU in the US (but this number might be different in the authors' calculations), this would mean that 50 million users receive over 75% exposure from like-minded sources. Compared to the aggregated average of viewers of cable news in prime time during 2020 (CNN, Fox News, and MSNBC), which is around 7 Million, I would not describe this as rare but actually quite a lot in absolute numbers.

The article presents good insights into the limitations of the analysis and what kind of studies are needed to overcome them. I could not agree more with the call in the article for replications in other countries with different political systems and over longer time periods in the future. However, I have doubts that such replications are possible, especially since they need to rely on Meta's automated estimation of the ideology of users based on user data that might not be publicly available (location, engagement with pages). While I do not doubt that this was possible in the US back in 2020-2021, both regulation and functionalities of Facebook have changed and I am not sure that replication is possible, especially in the EU. The GDPR is pretty clear about the need for approval when estimating sensitive attributes from user data, where ideology is explicitly mentioned as a sensitive attribute. This is a different case than what we are used to in previous articles where estimates were based on self-reported ideology or on estimates based on completely public data such as the following list of Twitter users. Also, ads targeting by ideology have been overall disabled on Facebook, which also makes me doubt whether a study like this one would be possible in the future even in the US. This does not subtract any quality or reliability from the results presented in the paper but, if my doubts are right, it is an important caveat regarding these needed studies in other countries and in future time periods. The authors, especially those working at Meta, are in the perfect position to clarify the extent to which these methods can be applied in future studies, especially in other countries beyond the US.

The article performs a careful statistical analysis that takes False Discovery Rates into account with a nuanced assessment of the priority of hypotheses and research questions. This was indeed important given the large list of dependent variables, which could have underpowered even the largest studies. I must say, however, that affective polarization (H1) is to me the a priori most important outcome and, without knowing the results, this would have been the only outcome to measure if I had to choose only one. The PAPE is not significant even before correcting for FDR but the SAPE is significant before correction and other outcomes also have no significant PAPE but significant SAPE even after correction for FDR (RQ2e: Pol. conv. on FB were informative, RQ4a: In-party affinity). While I do not think that the authors should cherry-pick these as actual effects of the experiment, I wonder about the reasons for the results in these three variables. A heterogeneity of effects could have been one, but the later assessment of this possible heterogeneity clears that doubt. The only explanation I can think of is a loss of power when calculating PAPE but I suspect the authors might know better explanations why the experiment sample might have shown a response that is not to be expected in a representative sample.

The authors report the precision and recall of each classifier in the SI but the sample sizes of the validation datasets of each classifier are not always reported. Some of these classifiers might have some classes much smaller than others and for those cases, we especially need the sizes per class too (or even better, reporting confusion matrices with exact counts). The most important classifier in this study is the ideology classifier, which is validated against the FIES survey for individual users and against the DW-NOMINATE scores of US members of congress when measuring page audience ideology scores from aggregated user labels. The sample size for the DW-NOMINATE validation is reported but not for the FIES panelists, at least not on page S175 where I would expect to see a count of self-reported liberals, conservatives, and moderates as for other classifiers. This is important in this case due to the low recall (21%) of the classifier for users who self-report to be moderates, which could be a large class. When the model is applied to the user viewing some content, the two-class classification precision and recall are above 80%, but when applied to the ideology of users as sources (with the 0.4/0.6 thresholds), this low performance might make the classifier mislabel users that are in reality moderate sources. I think this grants at least a comment on the article, since the downstream use of the classification in the statistical analysis ignores classification error and this might not be negligible depending on the count of panelists that self-report being moderate.

To sum up, the question tackled by this study is of extreme importance and I appreciate the effort and care that authors have put into every step of their work. My questions and suggestions above might help to clarify the article and I think are important given the impact that this paper will have not only in regulation but also in public opinion. I am optimistic about the authors being able to incorporate them. Please do not take the length of my review as a negative sign about the article but as the total opposite. This is extremely important work and my aim is to help the authors improve the article as much as I can.

Referee #3 (Remarks to the Author):

In "Like-minded sources on Facebook: Prevalent but not polarizing," the authors report the results of two separate studies that are used to investigate the extent to which the Facebook newsfeed promotes content from like-minded sources (Friends, Pages, and Groups), and whether a large reduction of such content results in a meaningful decrease in political polarization. The study is a truly impressive undertaking by a highly competent team of researchers, and the kind of access that the research team had to both Facebook's data and experimental infrastructure is notable in and of itself, after years of a shaky relationship between the platform and academic social science.

Before getting to the specifics of the analysis, findings, and framing of the manuscript, the last of which I found to be incomplete and troubling at times, it is worth noting that publication of the piece would likely result in widespread media coverage, and that signaling a more collegial relationship between Facebook and academia is good for both future scholarship and for perceptions of a platform that, for an extended period of time, acted in an opaque and disinterested manner towards social scientific collaboration.

There is no doubt that the data which were analyzed and the experiment which was conducted are both highly novel. I am not aware of any previous work that systematically measures the prevalence of like-minded sources on Facebook at such scale, nor do I know of any prior work that experimentally manipulates the composition of these sources on the platform itself to investigate their effects on political polarization. The findings from both the observational and experimental studies are more-or-less summarized in the title. On average, content from like-minded sources makes up about half of what Facebook users see in their newsfeeds, and the remainder mostly originates from sources that were neither like-minded nor cross-cutting. Moreover, changing the composition of these sources for several months, so that users see much less content from like-minded sources, does not result in a significant decrease in polarization across 8 pre-registered and widely-used measures. Furthermore, there were no significant heterogeneous treatment effects across a host of factors that the authors explored.

According to the authors, "these findings challenge popular narratives blaming social media 'echo chambers' for the problems of contemporary American democracy." But do they? In particular, can decreasing the amount of like-minded content for a few months be seen as comparable to flooding users with like-minded content for many years? There is a potential asymmetry here that the authors largely ignore throughout the piece, even in the title – the authors do not find that content from like-minded sources is "not polarizing," but rather that decreasing content from like-minded sources is not de-polarizing. Asymmetries abound in phenomena that are of social scientific interest – for example, prices are known to increase quickly with input costs but decrease slowly as costs go down, the so-called 'rockets and feathers' phenomenon in economics. More directly to the point, social trust is something that can break down instantly but take many years to build. It is feasible, though in no way certain or known, that years of negative framing of out-partisan messages (by like-minded sources) degraded trust in the opposing party in a way that is not easily fixed by decreasing the prevalence of these messages for a few months. I believe a more accurate conclusion to reach from the findings of the experiment is not that like-minded sources do not polarize, but rather that de-polarization work is difficult and cannot be easily solved with simple censoring solutions.

Beyond this, the prevalence of content from like-minded sources is also worth highlighting. In the pre-analysis plan that the authors include in the appendix (S13), the authors note that much prior work has found that, "only relatively small segments of the population consume large amounts of news from politically congenial sources." Depending on how broadly we conceive of 'news' here (i.e., whether this is referring to political news specifically or social posts more broadly), the findings of the observational study would seem to suggest that indeed Facebook users do receive quite a lot of information from like-minded sources. Even if we were to take a firm stance on defining 'news' in the more limited and traditional way as newspaper articles, the reason for why users see few news articles from like-minded sources is because they see few news articles period, not because they see relatively little information from like-minded sources. Indeed, as the authors note in the piece, "even content that is not explicitly about politics can still communicate relevant cues." That the authors found a high level of content originating from like-minded sources seems worthwhile to highlight in the concluding sentences of the manuscript, alongside the experimental results.

Finally, the fact that the authors found that a relatively low percentage of content can be labeled as 'civic' or news also seems notable. Despite the previously-stated worry that non-political content can include political cues, if the proportion of civic/news content that Facebook users

received in 2020 was as low in previous years, this would be a decent argument for why information from like-minded sources may not be polarizing after all – that is, because it is more likely to be personal than political. However, we do not know these proportions in previous years, in large part because for many years Facebook operated as a closed system that provided limited access for social or political inquiry. Given the platform’s 2015 launch of ‘Instant Articles,’ it seems likely that the proportion of news that users received on the platform may have varied over the years. And this is precisely why it is so important to know more about what Facebook users were exposed to in years prior to 2020 (when the study in question took place). It seems premature to let Facebook off the hook by suggesting that the experiment implies like-minded content plays little role in polarization, given that the study was not conducted during these earlier years when an escalation in polarization took place, and that the study investigates not the effects of like-minded content on polarization but rather its removal on de-polarization.

Below I include a few other minor comments that might help the researchers as they revise the manuscript. Overall, I think that the study is important and worth publishing, but that the framing of the results should be revised to focus more on the findings that the studies substantiate (high prevalence of like-minded content, low prevalence of news, difficulty of de-polarization) rather than the larger claims about echo chambers that the experiment, as impressive as it is, cannot address.

Minor comments:

- I know that post-hoc sub-group analyses are passé, but I was really hoping for a heterogenous treatment effect surrounding age. It would seem that, to make a case that like-minded content is not polarizing, it would be helpful to analyze younger users (i.e., those who may have been less likely to engage with political content during the polarizing years of the 2010s).
- It is somewhat surprising that civic/news content is coming from non-partisan sources. It makes me wonder what the nature of that content is. Specifically, is it politically neutral content or does it have a clear political leaning? It could be worthwhile to differentiate between echo chambers in sources vs. echo chambers in content.
- I may have missed this, but to what population are the results of the experiment meant to generalize via the weights – is it American Facebook users or American voters? It could be worthwhile to report both weighted estimates.

Author Rebuttals to Initial Comments:

Response to Reviewer 1:

- 1. Is it possible to provide more information about the intervention in the main text - by what magnitude did the intervention *attempt* to decrease exposure to like-minded content?***

The reviewer asks about the magnitude of the intended decrease in exposure to content from like-minded sources. We have expanded SI Section S1.1 to add new details about the implementation of our demotion, including details of the three elements (inventory, posts, and relevancy score) that go into the feed ranking algorithm and how these apply to our demotion. Our intervention did not have a specific target for the intended reduction of views, but instead was implemented as a multiplicative demotion based on the relevancy score assigned to posts from like-minded sources. The following text explains the process in detail:

Our demotion was implemented by multiplying the relevancy score assigned to posts created by like-minded sources by a known factor, which determines the strength of the demotion. The intervention affected the content that respondents in the treatment group were exposed to from their friends, the Groups of which they were a member, and the Pages that they followed. Ads and in-feed recommendations to content from Groups of which they are not a member, and Pages that they do not follow, were not affected by this intervention. The News Feed of users who were assigned to the control condition did not change as a result of the experiment. For our experiment, after exploring different possible strengths, we selected the strongest demotion we tested, which corresponded to a multiplication factor of 0.05. The decision to implement this type of demotion, as opposed to an intervention in which all content from like-minded sources is hidden (i.e., a multiplication factor equal to 0), was driven by our intention to strike a balance between an intervention whose effects would be observable and one that would not result in high levels of attrition. In practice, a demotion using a multiplication factor of 0.05 does not lead to a 95% reduction in exposure to the content affected by the demotion because some users do not have other posts in inventory that could be shown to them with a relevancy score higher than that of the demoted content. In other words, the final position in which posts from like-minded sources were displayed to each user in the treatment group depended on their inventory. For example, users whose friends, Groups, and Pages were primarily like-minded did not have to scroll down as far before they encountered like-minded content compared to those who had more diverse inventory. In our intervention, we observed a reduction in views from like-minded sources of around 33% (see Table S37). Additional details on how the effect of this

demotion varied on characteristics such as pre-treatment exposure to like-minded sources are available in SI 3.1.

Overall, our goal was to balance the need to implement an intervention with observable effects with the concerns that making the demotion too strong would, for some users, lead to no or very limited content in their feed (therefore affecting user experience and the usage of the platform as well as potentially leading to attrition issues). In addition to the expanded section in the SI, we have also added language in the main text that more succinctly describes the process as well as directs readers to the SI for additional detail:

Content from like-minded sources was downranked using the largest possible demotion strength that a pre-test demonstrated would reduce exposure without making the News Feed nearly empty for some users, which would have interfered with usability and thus confounded our results; see SI Section S1.1.

- 2. The abstract says: “The intervention increases exposure to content from cross-cutting sources and decreases exposure to uncivil language”. This language, and similar language in the main body of the paper, is not clear on whether this is something the intervention was supposed to accomplish mechanistically, or if this is a substantive finding.***

The reviewer asks us to clarify whether two of our findings (the increase in exposure to cross-cutting sources and decrease in uncivil language) were a deliberate part of the intervention’s design or whether they are substantive findings. We clarify that both are substantive findings, not a mechanistic part of the intervention. The objective of the intervention was solely to reduce exposure to information from like-minded sources; the effect of this reduction on exposure to other sources of content and the types of language people see was unknown. We changed the wording in the abstract to make it more clear that this is a substantive finding rather than part of the intervention:

We find that the intervention increases exposure to content from cross-cutting sources and decreases exposure to uncivil language

In addition, we further explain this distinction in the introduction, where we lay out potential consequences of reducing exposure to congenial content on dynamic social media platforms:

Finally, reducing exposure to like-minded information may not lead to a corresponding increase in exposure to information from sources with different political leanings (which we refer to as cross-cutting) and could also have unintended consequences. Social media feeds are typically limited to content from accounts that users already follow, which include few that are cross-cutting and many that are non-political (20). As a result, reducing exposure to like-minded sources may increase the prevalence of content from sources that are politically neutral rather than uncongenial. Furthermore, if content from like-minded sources is systematically different (e.g., in its tone or topic), reducing

exposure to such content may also have other effects on the composition of social media feeds.

We also offer a clarification in the Design section:

We note three important features of the design of the intervention. First, the sole objective of the intervention was to reduce content from like-minded sources. It was not designed to directly alter any other aspect of the participants' feeds.

3. *The paper said all users were assigned political leaning scores. How is it possible for a user to have a political leaning score if the user never engaged with or posted political content?*

Thank you for this question! We are happy to clarify. As stated in our description of the ideology classifier (see SI Section S6.1), it uses features that include users' demographics, preferred language, location, and engagement with both political *and* non-political Pages and Groups. In other words, for users who never engage with or share political content, the classifier makes a prediction based on whether their demographic characteristics or the non-political Pages and Groups that they engage with are more similar to the corresponding characteristics and engagement of users who self-report being conservative on their Facebook profile or to those who self-report being liberal. Because non-political content, Groups, and Pages can be associated with ideology and carry cues about one's political leaning (as also noted on p. 6), this information is useful. Our validations further show that the classifier is highly correlated with users' self-reported political leanings despite variation in the availability of explicitly political signals from some users (see SI Section S2.1.1).

4. *“Despite the prevalence of like-minded sources in what people see on Facebook, extreme “echo chamber” patterns of exposure are relatively rare. Just 20.6% of Facebook users get over 75% of their exposures from like-minded sources.” Is 20.6% rare?*

The reviewer is of course correct that our characterization of the finding as “relatively rare” is inherently subjective. We characterized the finding in this way because our reading of the literature on “echo chambers” suggests that critics think the problem is pervasive and widespread. We wanted to highlight that a minority of users reside in extreme echo chambers. However, we acknowledge that priors may differ about the scale of the “echo chamber problem” and the set of comparisons one might select. In absolute terms, there is similarly no established benchmark for what constitutes an “echo chamber” given the differences in definitions, operationalizations, and measurements in past work. Finally, our measurement of content exposure from all like-minded sources (rather than just explicitly political content) makes this estimate difficult to compare with prior research. Given these considerations, we have changed our description from “relatively rare” to “infrequent” per the reviewer's question. We hope this terminology offers an acceptably neutral description, but are of course open to suggestions from the editor or reviewer about alternate phrasing.

- 5. Figure 3 – It is hard to interpret an effect on certain variables, eg “vote choice and candidate evaluation” – the reader needs to know what the variable is, and what higher v lower means. eg For figure C is “active engagement” the rate or total? I recommend adding full variable definitions to the caption.**

We appreciate the reviewer’s point. Given the large number of outcome measures, we lacked space to add full variable definitions to the caption, but the reviewer is of course correct that we should ensure the reader is provided with the information that is required to correctly interpret our findings. We have therefore revised the figure as follows. First, we added horizontal lines and colons for the grouping labels of the outcome variables to make more clear that the non-bolded labels underneath a bolded label fall into that category. We have also added corresponding text to the caption to aid interpretation: “Non-bolded outcomes that appear below a bolded header are part of that category. For example, in Panel D, ‘Issue positions,’ ‘Group evaluations,’ and ‘Vote choice and candidate evaluations’ all appear below ‘Ideologically consistent views,’ indicating that all are measured such that higher values indicate greater ideological consistency.” Finally, we have revised the caption to point the reader directly to the section of the SI where we provide the full question wording and variable construction for each outcome measure: “Full descriptions of all outcome variables can be found in SI S1.4.”

To answer the reviewer’s specific question, Panel C is titled “Engagement Rate,” which we report to contrast with the measures of “Total Engagement” reported in Panel B. The caption describes this difference in this way: “Panel B specifically reports total engagement (for content, the total number of engagement actions) while Panel C reports the engagement rate (the probability of engaging conditional on exposure).” Thus, “active engagement” in Panel C is the rate while “active engagement” in Panel B is the total. If the reviewer has a suggestion about how to relabel Panels B and C to make that more clear, we are of course open to suggestions.

- 6. A number of concepts are used in the paper that are not defined in the main body of the paper. I imagine the authors are cognizant of space constraints, but I think it’s important that these definitions are pushed into the main text. One example: “cross-cutting sources”. Another example is that the introduction says: “First, the majority of the content that active adult Facebook users in the U.S. see comes from like-minded friends, Pages, and Groups, although only small fractions of this content are categorized as news or explicitly about politics.” I believe like-minded is defined vis a vis politics later in the paper, but I think it would be clearer if it was defined on first use. I don’t believe like-minded is ever defined for non-political issues/topics.**

This is an excellent point and we have addressed it in two ways. First, we added brief definitions of key concepts within the introduction. We define like-minded sources in the first paragraph here:

To assess how often people are exposed to congenial content on social media, we use data from all active adult U.S. Facebook users to analyze how much of what they see is from sources that we categorize as sharing their political leanings (which we refer to as content from like-minded sources; see Design below).

We then define cross-cutting sources when we introduce the concept a few paragraphs later:

Finally, reducing exposure to like-minded information may not lead to a corresponding increase in exposure to information from sources with different political leanings (which we refer to as cross-cutting) and could also have unintended consequences.

In addition, in the Design section, we attempted to better clarify how we operationalize “content from like-minded sources,” emphasizing that this category is not restricted only to news or political information, but rather to *any* information from friends, Pages, and Groups who share a participant’s political leaning:

The News Feed of participants in the control condition was not systematically altered. Due to the difficulty of measuring the political leaning or slant of many different types of content at scale, we instead varied exposure to content based on the estimated political leaning of the source of the information. Using a Facebook classifier, we estimate the political leaning of other users directly (see SI Section S1.3 for details). Building on prior research (14,15,21,33,34), we estimate the political leanings of Pages and Groups using the political leanings of their audience (e.g., Group members and Page followers). For participants assigned to treatment, we downranked all content (including, but not limited to, civic and news content) from friends, Groups, and Pages who were predicted to share the participant’s political leaning using this approach (e.g., all content from conservative friends as well as Groups and Pages with conservative audiences was downranked for participants classified as conservative; see SI Section S1.1).

Response to Reviewer 2:

- 1. Among some suggestions for improvement of the article, I think the extent of the effect of the treatment in the exposure to like-minded sources is hard to assess in the current manuscript. It takes until the end of page 3 to see that the reduction of exposure to like-minded sources was for about one-third, this could be mentioned more clearly in the abstract and at times when describing the intervention. The reader needs to be aware that the downranking of like-minded sources was not complete and still users saw more like-minded content than cross-cutting content (36.2% vs 27.9%) and whether this differs from what we would expect from a plain chronological ranking based on friends. I suspect that this "no algorithm" case would still be very skewed toward like-minded sources and the actual effect of the algorithm change is stronger than having no algorithm but still less than showing a random sample from all of US Facebook.***

We agree with the suggestion to describe the magnitude of the intervention earlier on in the paper and have added this language in the introduction (text in italics added):

Second, we find that an experimental intervention reducing exposure to content from like-minded sources *by about a third* reduces total engagement with that content and decreases exposure to content classified as uncivil and content from sources that repeatedly post misinformation.

In addition, as also suggested by reviewer 1, we describe the intended strength of the intervention to make the reader aware that the downranking was not – and could not – be complete (see the revised version of SI Section S1.1). On the one hand, we wanted to make the intervention as strong as possible. On the other hand, making the demotion too strong would leave no or very limited content in the feed of some users (therefore affecting user experience and the usage of the platform, as well as potentially leading to attrition issues). This point relates to the reviewer’s observation that users still saw more like-minded content than cross-cutting content: inasmuch as some users have very little content from cross-cutting sources in their inventory (e.g., because they are not Facebook friends with individuals, followers of Pages, or members of Groups that are politically uncongenial), demoting like-minded sources would not necessarily fill the void with cross-cutting sources.

On that note, we have added language through the paper to clarify how cross-cutting content fits into our experimental design. To summarize, increasing exposure to cross-cutting content was not a deliberate goal of the experiment. Rather, one of the questions we sought to investigate was how our treatment would change the composition of content in participants’ News Feeds – and in particular, the relative increases of content from cross-cutting sources versus content from sources that are neither like-minded or cross-cutting (which will depend in part on the participants’ networks).

[R]educing exposure to like-minded information may not lead to a corresponding increase in exposure to information from sources with different political leanings (which we refer to as cross-cutting) and could also have unintended consequences. Social media feeds are typically limited to content from accounts that users already follow, which include few that are cross-cutting and many that are non-political (20). As a result, reducing exposure to like-minded sources may increase the prevalence of content from sources that are politically neutral rather than uncongenial. Furthermore, if content from like-minded sources is systematically different (e.g., in its tone or topic), reducing exposure to such content may alter patterns of exposure in other ways as well.

Finally, we agree that comparing the effect of our treatment to the effect of the algorithm would be an excellent direction for future research. The reviewer’s intuition about the distribution of content from like-minded sources in a reverse chronological feed strikes us as correct, especially given that people do have more content from like-minded friends, Groups and Pages in their inventory than from cross-cutting friends. However, we cannot speak to this in the

manuscript because our project was designed to investigate the effects of reducing like-minded exposure on the Facebook platform as it currently exists today, which includes the algorithmic ranking of the News Feed.

- 2. A change that would help in this respect is to show in Figure 2 the mean day-level share of respondent views of content from cross-cutting sources, perhaps with another color, so that the reader can see how that exposure has changed too and to which extent it stays below the exposure to like-minded sources even after the beginning of the treatment.***

We agree with the reviewer that it would be helpful to visualize the effect of the treatment on the day-level share of respondent views from cross-cutting sources. We have made such a visualization and included it in SI Section 3.1. We also alert the reader to its existence in the caption of Figure 2 in the main text. We chose not to alter Figure 2 directly because we wanted to ensure that readers understand the nature of the intervention itself: a demotion of content from like-minded sources in users' News Feeds. While we anticipated that this intervention might simultaneously result in an increased proportion of content from cross-cutting sources, we did not directly intervene on that aspect of the Feed. As suggested by the reviewer, we now also comment on this explicitly in SI Section S3.1:

The treatment increased exposure to content from cross-cutting sources relative to the pre-treatment period. The magnitude of this increase was not equivalent to the decrease in exposure to like-minded sources because exposure to content from sources that were neither like-minded nor cross-cutting also increased. However, exposure to cross-cutting sources remained lower than exposure to like-minded sources throughout the entire study period.

3. *The observational study of the full sample of Facebook users is impressive and something that seizes the best opportunities of a collaboration like this one. However, the size of this full sample is never disclosed in the article. Even table S3, which directly compares the survey respondents and the US MAP sample, reports the exact N of each calculation on the survey sample but the N for US MAP is always reported as "100%". To assess the results of the full sample, we need to know the number of users included in it (at least approximately) and more descriptive statistics, especially about ideology scores. To interpret the precision and recall values of the ideology classifier, we need to know the sample size of users labeled as each, or even better: the histogram of ideology scores. This will also allow a reader to see whether the 0.5 threshold could be biased or how sensitive the distribution is to the 0.4/0.6 thresholds used later. A report on the full sample size would also help to understand the fraction of the US population that is included in this observational study. The article makes a great motivation point when citing that Twitter is only used by 23% of the public, but with the data reported here, we should have a direct number of the percentage of the above-18 public in the US that is included in the study.*

I acknowledge that this kind of request might be hard to fulfill in an industry-academia collaboration and the last thing I want is to delay the article with unreasonable requests but I think that sample size is an important metric to report for research transparency, also for observational digital-trace data analysis. I think that the Meta/academic agreement would allow this kind of reporting of aggregate numbers that bring no risk to privacy. This might be spelled out more in

detail in the linked OSF repository (<https://osf.io/7wpgd>) but it was not online at the time of my review and I have the impression that this would not require any review by Meta since reporting these numbers are not an issue regarding confidential or personally identifiable information or to play a role in abiding by existing legal obligations.

We think the reviewer’s request is quite reasonable and have worked to implement it in several ways.

The reviewer asked for more information about the size of the US MAP population. We have updated the main text and SI to clarify that our analysis of monthly active users represents the subset of adult (18+) users among the 231 million users who accessed Facebook every month during our study period. We also added this information to S2.1.1 to aid readers in interpreting the tables provided in that section that compare our participants to MAP. For example, a reader could use this figure to help interpret Table S3 in the SI, which includes the distribution of the predicted ideology score among US MAP.

At the reviewer’s request, we have also provided more information about the performance of the ideology classifier among our participants, leveraging the self-reported information they provided in our surveys for validation purposes. First, we add considerably more detail in section S2.1.1 of the SI about how we validated the ideology classifier. These details include classifier confusion matrices that visualize the recall and accuracy of the measure based on self-report (including the sample size in each subcategory), plots showing the distribution of self-reported ideology within bins of size 0.10 of the (0,1) classifier estimates, and as requested, histograms of the distribution of the classifier measure within the sample for subgroup of participants as defined by their ideology, partisan identification, and Trump approval. We provide these histograms below as an illustration of the new analyses we conducted to incorporate the reviewer’s suggestion.

Second, we have updated the header of section S3.11 in the supplementary information to flag what we see as an important robustness check to potential concerns related to our usage of the ideology classifier. This analysis shows that our main conclusions remain identical if we exclude users with an ideology score in the (0.4 to 0.6) range, where our classifier performs less well.

- 4. On a related comment, the paper is rather clear about Meta not having pre-publication approval and only being entitled to review the paper regarding these sensitive privacy and legal reasons, but what remains unclear in the article is if Meta did in fact review the article beyond the authors listed in the study. The separation between changes requested for research or for legal/privacy obligations is blurry and, while it's clear that the academic authors here did their best and the process was a great example of transparency, a reader needs to know more of the extent to which Meta did take a role in reviewing the content of the paper and not that it was just a possibility.***

We agree that this is an important consideration and created several safeguards to prevent any undue influence from Meta on the research findings beyond its basic legal/privacy obligations. First, by uploading a detailed pre-registration to OSF prior to data collection, we bound ourselves (and Meta) to reporting all results no matter how they reflected on the company. Second, the project has an independent rapporteur who participated in many of the meetings between the academics and Meta and will be publishing their observations in a future article.

The purpose of the review by the Meta's Legal/Privacy department was solely to ascertain that the main article and the supplementary materials did not contain any confidential data shared with the academic researchers as part of this project or data that could reveal users' personally identifiable information as well as to ensure that Meta was abiding by their existing legal obligations. Most relevant to the reviewer's comment, we confirm that Legal/Privacy did not request any changes following the pre-publication review of this paper, which we now note in Section S11 of the SI. In fact, as we note in S11 as well, Meta publicly agreed that there would be no pre-publication approval of papers for publication on the basis of their findings.

Finally, the draft of this paper was only accessible to the Meta research team involved in this project and the legal and privacy reviewers; it was not shared with anyone else at the company. We also make these points clear in the response to the Editor at the beginning of this memo. Thank you for this important query, which is key to providing necessary transparency regarding Meta's oversight of the paper.

- 5. The reporting of sample sizes in the observational study is also important to contextualize the finding that just 20.6% of Facebook users get over 75% of their exposures from like-minded sources, which is discussed in the article as evidence that is not consistent with the worst fears about echo chambers. While this is not as bad as what is known from offline interaction nor as prevalent as how the echo chamber has been portrayed, the picture is different if we think about Facebook as***

news media. Assuming that Facebook had about 250 Million MAU in the US (but this number might be different in the authors' calculations), this would mean that 50 million users receive over 75% exposure from like-minded sources. Compared to the aggregated average of viewers of cable news in prime time during 2020 (CNN, Fox News, and MSNBC), which is around 7 Million, I would not describe this as rare but actually quite a lot in absolute numbers.

The reviewer raises a good point: even if only a minority of people receive most of their content from like-minded sources, Facebook's reach means that this group is still large in absolute terms. We address this comment in three ways. First, we previously characterized the incidence of this group as "relatively rare," a subjective term that depends on the priors one holds about the scale of the "echo chamber problem" and the comparisons one might make to other phenomena. We thus changed the terminology from "relatively rare" to "infrequent," which we hope is a more neutral description, but we are of course open to suggestions from the editor or reviewer about alternate phrasing. Second, the reviewer suggests thinking about Facebook as news media and mentions cable news viewership in prime time. This comparison is tricky, however, given that we measured all exposure to content from like-minded sources, which includes content from like-minded friends, Groups, and Pages that is ostensibly non-political as well as the explicit news and/or political content that has been the focus of past studies. As such, our estimates of congenial exposure are difficult to compare with prior research.

To address the reviewer's concerns, we note these points explicitly in the conclusion, reminding readers of the size of the "tail" of the distribution in absolute terms and encouraging more research into the causes and consequences of highly skewed patterns of information consumption:

In addition, though only a minority of Facebook users occupy "echo chambers," the reach of the platform means that the group in question is large in absolute terms. Future research should seek to better understand why some people are exposed to large quantities of like-minded information and the consequences of this exposure.

- 6. The article presents good insights into the limitations of the analysis and what kind of studies are needed to overcome them. I could not agree more with the call in the article for replications in other countries with different political systems and over longer time periods in the future. However, I have doubts that such replications are possible, especially since they need to rely on Meta's automated estimation of the ideology of users based on user data that might not be publicly available (location, engagement with pages). While I do not doubt that this was possible in the US back in 2020-2021, both regulation and functionalities of Facebook have changed and I am not sure that replication is possible, especially in the EU. The GDPR is pretty clear about the need for approval when estimating sensitive attributes from user data, where ideology is explicitly mentioned as a sensitive attribute. This is a different case than what we are used to in previous articles where estimates were based on self-reported ideology or on estimates**

based on completely public data such as the following list of Twitter users. Also, ads targeting by ideology have been overall disabled on Facebook, which also makes me doubt whether a study like this one would be possible in the future even in the US. This does not subtract any quality or reliability from the results presented in the paper but, if my doubts are right, it is an important caveat regarding these needed studies in other countries and in future time periods. The authors, especially those working at Meta, are in the perfect position to clarify the extent to which these methods can be applied in future studies, especially in other countries beyond the US.

The reviewer is correct that Meta's ability to engage in similar collaborations with academic researchers internationally or in the US may be constrained by the broader regulatory environment both in the United States and abroad. Due to space limitations and because this issue is beyond the control of the authors, Meta, and other researchers, the text in the discussion now states that further research should be conducted "wherever possible":

Finally, replications in other countries with different political systems and information environments will be essential to conduct wherever possible to determine how these results generalize.

Section S11 of the SI further discusses these important caveats when we call for more comparative work of this kind:

We acknowledge the importance of such transparent collaborations with academic researchers internationally and in the United States beyond the scope of this project. However, we also recognize that Meta's ability to engage in similar collaborations may be constrained by the broader regulatory environment. Given that both regulations and functionalities of Facebook change rapidly (e.g., the GDPR in the European Union or the disabling of ads targeting by ideology), replications and extensions of this project may be challenging (if not impossible in some places).

The Meta employees who are co-authors on this paper are not in a position to speak for the company about the possibility of further studies in the US or in other countries, but we note that the datasets used to conduct this study will be accessible to researchers through ICPSR to enable replication and additional studies investigating Facebook use during elections.

7. The article performs a careful statistical analysis that takes False Discovery Rates into account with a nuanced assessment of the priority of hypotheses and research questions. This was indeed important given the large list of dependent variables, which could have underpowered even the largest studies. I must say, however, that affective polarization (H1) is to me the a priori most important outcome and, without knowing the results, this would have been the only outcome to measure if I had to choose only one. The PAPE is not significant even before correcting for FDR but the SAPE is significant before correction and other

outcomes also have no significant PAPE but significant SAPE even after correction for FDR (RQ2e: Pol. conv. on FB were informative, RQ4a: In-party affinity). While I do not think that the authors should cherry-pick these as actual effects of the experiment, I wonder about the reasons for the results in these three variables. A heterogeneity of effects could have been one, but the later assessment of this possible heterogeneity clears that doubt. The only explanation I can think of is a loss of power when calculating PAPE but I suspect the authors might know better explanations why the experiment sample might have shown a response that is not to be expected in a representative sample.

The reviewer correctly identifies some discrepancies between the PATE and SATE estimates. Our preregistration committed us to reporting both quantities and to using particular standards for evaluating the significance of individual statistical tests. Absent preregistered analyses explicitly designed to address these differences, we don't speculate about the reasons for their existence. However, for readers who are interested in hypothesizing about how the differences in the characteristics of our sample might contribute to the SATE/PATE discrepancies, we provide information that we hope will be helpful. We include an extensive comparison between our experimental sample and more representative samples in Table S2 in the supplemental information. Our pre-registered HTE analyses are also designed to assess whether variation in particular characteristics made some users more responsive to the treatment than others.

- 8. The authors report the precision and recall of each classifier in the SI but the sample sizes of the validation datasets of each classifier are not always reported. Some of these classifiers might have some classes much smaller than others and for those cases, we especially need the sizes per class too (or even better, reporting confusion matrices with exact counts). The most important classifier in this study is the ideology classifier, which is validated against the FIES survey for individual users and against the DW-NOMINATE scores of US members of congress when measuring page audience ideology scores from aggregated user labels. The sample size for the DW-NOMINATE validation is reported but not for the FIES panelists, at least not on page S175 where I would expect to see a count of self-reported liberals, conservatives, and moderates as for other classifiers. This is important in this case due to the low recall (21%) of the classifier for users who self-report to be moderates, which could be a large class. When the model is applied to the user viewing some content, the two-class classification precision and recall are above 80%, but when applied to the ideology of users as sources (with the 0.4/0.6 thresholds), this low performance might make the classifier mislabel users that are in reality moderate sources. I think this grants at least a comment on the article, since the downstream use of the classification in the statistical analysis ignores classification error and this might not be negligible depending on the count of panelists that self-report being moderate.**

Thank you for these important points and the close reading of the details on the classifiers. First, regarding classifiers more generally: the reviewer is correct in pointing out that all the classifiers

we rely on (with the important exception of ideology) suffer from a class imbalance problem since they are designed to identify subsets of content on Facebook that likely do not represent over 50% of all posts. For this reason, we note that all precision and recall statistics reported in Section S6 of the SI are always in reference to the less frequent category for each classifier. For example, for the civic classifier, the 83% precision and 82% recall statistics on English-language Facebook content refer to this classifier's ability to predict whether a post is civic (and not whether it is *not* civic).

Second, in response to the reviewer's suggestion to add more information on how the ideology classifier was validated, we have now expanded SI Section S2.1.1 to add three additional analyses: (1) graphs with confusion matrices for the two-category and three-category prediction tasks, which include the sample sizes of each subgroup, (2) stacked bar charts of the distribution of self-reported ideology at different values of the ideology scores; and (3) histograms with the distribution of ideology scores for each subgroup of self-reported ideology, party identification, and Trump approval in the survey data.

The reviewer expressed concern that the recall statistic for the "moderate" group in the classifier seemed low. We hope that these additional analyses (and in particular, the accuracy of the classifier with regard to party identification and Trump support) help illustrate why the tendency of the classifier to predict self-reported moderates to be left- or right-leaning should not be a major concern for our analyses. First, this pattern is consistent with external research finding that self-reported ideology is an imperfect measure: self-reported moderates often have non-centrist ideological positions on various issues (see e.g. Broockman, 2016). One of the new figures that we added to SI Section S2.1.1, which we reproduce below, demonstrates a similar pattern: a large number of participants who self-report having "middle-of-the-road" ideological views do not have ideology scores in the "moderate" category (from 0.4 to 0.6), identify as Democrat or Republican, and either approve or disapprove of Trump's performance as president. The implication of this pattern for our analysis is that most users who self-reported being moderate had political leanings that in practice caused them to be correctly classified on one side or the other of the political spectrum.

Due to the nature of this project, we unfortunately do not have additional information on the sample size and distribution of labels in classifiers provided by Meta beyond what is currently in SI Section S6. Meta was able to share details about the coverage and performance of its internal classifiers, but was not able to provide additional information beyond that because the classifiers use proprietary methodologies. As a result, it is not possible to share this information within the parameters of this collaboration.

The trade-offs associated with using classifiers that are only available from Meta is an issue that we discussed extensively at the design stage of this project. For this reason, we explored using open-sourced methods from peer-reviewed published research that could be implemented on the data provided by Meta and could perform well. In some cases, appropriate open-source classifiers were available (for example, when identifying uncivil or hateful content). In the case of political or news content, however, this proved to be infeasible due to the lack of a comparable open-source classifier for Facebook posts that could be validated at scale for all sources and the time constraints under which we launched the project, which limited our ability to develop new classifiers.

Response to Reviewer 3:

1. ***According to the authors, “these findings challenge popular narratives blaming social media ‘echo chambers’ for the problems of contemporary American democracy.” But do they? In particular, can decreasing the amount of like-minded content for a few months be seen as comparable to flooding users with like-minded content for many years? There is a potential asymmetry here that the authors largely ignore throughout the piece, even in the title – the authors do not find that content from like-minded sources is “not polarizing,” but rather that decreasing content from like-minded sources is not de-polarizing. Asymmetries abound in phenomena that are of social scientific interest – for example, prices***

are known to increase quickly with input costs but decrease slowly as costs go down, the so-called 'rockets and feathers' phenomenon in economics. More directly to the point, social trust is something that can break down instantly but take many years to build. It is feasible, though in no way certain or known, that years of negative framing of out-partisan messages (by like-minded sources) degraded trust in the opposing party in a way that is not easily fixed by decreasing the prevalence of these messages for a few months. I believe a more accurate conclusion to reach from the findings of the experiment is not that like-minded sources do not polarize, but rather that de-polarization work is difficult and cannot be easily solved with simple censoring solutions.

We thank the reviewer for this insightful point and appreciate the encouragement to think about our results in this framework. Existing theory suggests that exposure to like-minded content has the potential to polarize users. Therefore, for ethical reasons, we chose NOT to design a platform intervention that would directly test this proposed polarizing mechanism and instead to test an intervention that had the potential to *depolarize*. The reviewer is entirely correct that these effects may not be symmetric. In other words, the polarizing and depolarizing effects of exposure to content from like-minded sources may not be equivalent, especially given that the potential differences in treatment dosage (decreasing the amount of content from like-minded sources for a few months vs. cumulative exposures to such content on social media platforms and elsewhere for many years). To address this important point, we expand two sections of the paper. First, in the Design section, we describe that we chose to test the effects of like-minded information by reducing rather than increasing exposure:

[G]iven the associations between polarized attitudes and exposure to politically congenial content that have been found in prior research, we deliberately designed an intervention that reduces rather than increases exposure to content from like-minded sources to minimize ethical concerns.

The second is in the Discussion section, where we raise the asymmetry point to contextualize our null effects:

Finally, we sought to decrease rather than increase exposure to like-minded information for ethical reasons. While the results suggest that decreasing exposure to information from like-minded sources has minimal effects on attitudes, the effects of such exposure may not be symmetrical. Specifically, decreasing exposure to like-minded sources might not reduce polarization as much as increasing exposure would exacerbate it.

Given these concerns about asymmetry, we have also revised the paper to avoid claiming that the intervention tests the effects of “echo chambers” -- see our response to point 4 below.

Lastly, the reviewer also mentioned the related issue of potential cumulative effects, stating that decreasing the prevalence of messages from like-minded sources for a few months would not

easily fix “years of negative framing of out-partisan messages (by like-minded sources).” We address this issue in responding to points 3 and 5 below.

- 2. Beyond this, the prevalence of content from like-minded sources is also worth highlighting. In the pre-analysis plan that the authors include in the appendix (S13), the authors note that much prior work has found that, “only relatively small segments of the population consume large amounts of news from politically congenial sources.” Depending on how broadly we conceive of ‘news’ here (i.e., whether this is referring to political news specifically or social posts more broadly), the findings of the observational study would seem to suggest that indeed Facebook users do receive quite a lot of information from like-minded sources. Even if we were to take a firm stance on defining ‘news’ in the more limited and traditional way as newspaper articles, the reason for why users see few news articles from like-minded sources is because they see few news articles period, not because they see relatively little information from like-minded sources. Indeed, as the authors note in the piece, “even content that is not explicitly about politics can still communicate relevant cues.” That the authors found a high level of content originating from like-minded sources seems worthwhile to highlight in the concluding sentences of the manuscript, alongside the experimental results.***

The reviewer brings up an interesting point. Per the reviewer, exposure to news online in general is relatively rare (e.g., Guess, 2021; Wojcieszak et al., 2021); this is also true in our data (a median of 6.7% of what participants saw was classified as news). As the reviewer notes, however, our finding that “the median Facebook user received a majority of their content from like-minded sources” appears to contrast with prior research showing that heavily slanted online information diets are rare (e.g., Grinberg et al. 2020, Guess 2021). While there are many important differences between the design and measurement approach in our study and prior studies like those cited above (e.g., relying on online browsing data from desktop users and focusing on news websites), we very much wish to acknowledge the point. We have therefore revised a passage in the conclusion to draw more attention to this fact and to remind people about the potential of like-minded non-political content to reinforce people’s political attitudes:

We provide the most systematic descriptive evidence to date of the extent to which social media users disproportionately consume content from politically congenial sources. On the one hand, only a small proportion of the content that Facebook users see explicitly concerns politics or news and relatively few users have extremely high levels of exposure from like-minded sources. On the other hand, a majority of the content that active adult Facebook users in the U.S. see on the platform comes from politically like-minded friends or from Pages and Groups with like-minded audiences (mirroring patterns of homophily in real-world networks (13, 38)). This content has the potential to reinforce partisan identity even if it is not explicitly political (12).

- 3. Finally, the fact that the authors found that a relatively low percentage of content can be labeled as ‘civic’ or news also seems notable. Despite the previously-***

stated worry that non-political content can include political cues, if the proportion of civic/news content that Facebook users received in 2020 was as low in previous years, this would be a decent argument for why information from like-minded sources may not be polarizing after all – that is, because it is more likely to be personal than political. However, we do not know these proportions in previous years, in large part because for many years Facebook operated as a closed system that provided limited access for social or political inquiry. Given the platform’s 2015 launch of ‘Instant Articles,’ it seems likely that the proportion of news that users received on the platform may have varied over the years. And this is precisely why it is so important to know more about what Facebook users were exposed to in years prior to 2020 (when the study in question took place). It seems premature to let Facebook off the hook by suggesting that the experiment implies like-minded content plays little role in polarization, given that the study was not conducted during these earlier years when an escalation in polarization took place, and that the study investigates not the effects of like-minded content on polarization but rather its removal on de-polarization.

We certainly agree that our experiment does not provide conclusive evidence that Facebook has *not* contributed to polarization. As the reviewer points out, some of its effects may have occurred years ago during a period in which the user population, political context, and platform itself were all different than they were in 2020. While our experiment cannot address these issues directly, we conducted exploratory analyses to test two possible observable implications of these points that are relevant to our experimental findings. Specifically, in response to both this comment and the comment 5 below, we tested whether treatment effects on the outcomes for our primary hypotheses varied by respondent age or years since joining the platform. If the effects of online or social media information exposure on political attitudes are strongest when people are first exposed, then we might expect that younger people and those who joined the platform more recently would be more strongly affected by our intervention. We find no evidence that this is the case, however. We include these results in SI Section S3.10.5 and explicitly mention them at the end of the results section as well as the discussion section, where we mention these findings in connection with the question about the effects of prior Facebook use:

We note several other areas for future research. First, we cannot rule out the many ways in which social media use may have affected participants' beliefs and attitudes prior to the experiment. In particular, our design cannot capture the effects of prior Facebook use or cumulative effects over years; experiments conducted over longer periods and/or among new users are needed (although we find no evidence of heterogeneous effects by age or years since joining Facebook).

It is important to be clear that we recognize the heterogeneous treatment results discussed above are not definitive. We report them because they help us speak to the question at hand to the extent possible given the available data but agree with the reviewer about the temporal scope of our findings. We hope that future research will further investigate these questions.

(Note: These findings are also discussed below in relation to R3's point 5.)

- 4. Below I include a few other minor comments that might help the researchers as they revise the manuscript. Overall, I think that the study is important and worth publishing, but that the framing of the results should be revised to focus more on the findings that the studies substantiate (high prevalence of like-minded content, low prevalence of news, difficulty of de-polarization) rather than the larger claims about echo chambers that the experiment, as impressive as it is, cannot address.**

We agree with the Reviewer that it is important to describe what we found as carefully and specifically as possible. We have therefore rephrased several portions of the manuscript to emphasize the points highlighted by the reviewer and to reduce or eliminate references to “echo chambers” when describing the experiment per the reviewer’s concern. For example, the abstract has been amended to describe the experiment as “evaluat[ing] a potential response to concerns about the effects of ‘echo chambers’” rather than “identify[ing] the causal effects of ‘echo chambers’ on political attitudes.” We also rephrased the second paragraph of the introduction to state more precisely that we examine “the frequency with which people are exposed to congenial content” and to describe our intervention as “a potential response to concerns about the effects of ‘echo chambers.’” In the design section, we further clarify that our design “reduces rather than increases exposure to content from like-minded sources to minimize ethical concerns.” We also removed the only reference to “echo chambers” in the results section and removed the sentence evaluating our findings in the context of the “echo chamber concern” from the conclusion. In total, the manuscript now references the term “echo chambers” to motivate the research (given that the media characterization of and public debate about information exposure on social media still heavily rely on this term) and considers the descriptive findings in that context, but interprets the experiment as a potential response to concerns about “echo chambers” rather than a test of their effects. We hope these revisions satisfy the reviewer’s concerns but welcome further feedback on these points.

- 5. I know that post-hoc sub-group analyses are passé, but I was really hoping for a heterogenous treatment effect surrounding age. It would seem that, to make a case that like-minded content is not polarizing, it would be helpful to analyze younger users (i.e., those who may have been less likely to engage with political content during the polarizing years of the 2010s).**

We appreciate the reviewer’s point; many people wonder about whether long-term exposure to polarized sources online creates durable effects that are hard to undo. We have therefore undertaken two exploratory analyses to address this point that are separate from the preregistered heterogeneous treatment effect analysis reported in the paper. The first tests for heterogeneous effects by age group, allowing us to test per the reviewer’s suggestion if younger people are more responsive to changes in exposure to like-minded sources. The second analysis, which we believe is complementary, tests for heterogeneous effects by quintile of tenure --- i.e., the number of years since a participant joined Facebook.

The results, which are reported in Section S3.10.5, find no evidence of heterogeneous treatment effects on our primary outcome measures. We summarize these findings in the main text as follows. First, we briefly note the null effect in the results section:

Similarly, an exploratory analysis finds no evidence of heterogeneous effects by age or years since joining Facebook.

The full set of results are reported in SI Section S3.10.5, including figures. As we note there, none of the subgroup effects are significant after a false discovery rate adjustment. In particular, there is no clear evidence of larger or different effects among people who are younger or have been on Facebook for the least amount of time.

Second, we note these findings in the conclusion as relevant to concerns about long-term effects of Facebook use:

In particular, our design cannot capture the effects of prior Facebook use or cumulative effects over years; experiments conducted over longer periods and/or among new users are needed (though we find no evidence of heterogeneous effects by age or years since joining Facebook).

(Note: These findings are also discussed above in relation to R3's point 3.)

6. *It is somewhat surprising that civic/news content is coming from non-partisan sources. It makes me wonder what the nature of that content is. Specifically, is it politically neutral content or does it have a clear political leaning? It could be worthwhile to differentiate between echo chambers in sources vs. echo chambers in content.*

The reviewer raises an important point. As a reminder, we partition sources into those that are “like-minded” to respondents (i.e., on the same side of the political divide), those that are “cross-cutting” (i.e., those that are on the opposite side), and those that are classified as neither like-minded nor cross-cutting. The last category does not necessarily mean the sources are non-partisan; per our measurement strategy, these are simply friends, Pages, or Groups that have a score between 0.4 and 0.6 using the internal Facebook classifier to estimate their political leanings:

We used an internal Facebook classifier to estimate the political leaning of adult Facebook users in the U.S. (see SI Section S6 for classifier details and validation). The classifier produces predictions at the user level ranging from 0 (left-leaning) to 1 (right-leaning). Users with predicted values greater than .5 were classified as conservative and otherwise classified as liberal, allowing us to analyze the full population of active adult Facebook users in the U.S. A Page's score is the mean of the users who follow the page and/or share its content; a Group's score is the mean of Group members and/or users who share its content. We classified friends, Pages, or Groups as liberal if their predicted

value was 0.4 or below and conservative if it was 0.6 or above. This approach allows us to identify sources that were clearly like-minded or cross-cutting with respect to users (friends, Pages, and Groups with values between 0.4 and 0.6 were treated as neither like-minded nor cross-cutting).

There are thus a number of ways that a source could be classified as neither like-minded nor cross-cutting. For instance, the set of possibilities includes all of the following:

- Pages and Groups with an audience (those who follow the Page or are a member of a Group and/or share its content) that is politically divided but relatively balanced;
- Pages and Groups with largely moderate audiences;
- Friends who are estimated to have moderate or centrist leanings.

These sources may share content that is classified as civic (i.e., about politics and social issues) and/or news. We anticipate that such content is heterogeneous and includes a mix of politically neutral content and content with a clear political leaning.

Per the reviewer's final point, we want to be as clear as possible. In the absence of a validated classifier of content-level slant that could be applied at scale to many types of content (videos, images, text, etc.), we are unable to estimate the slant of the content that participants see or to intervene on it based on its estimated slant. To clarify this point for readers, we added this sentence to the design section:

Due to the difficulty of measuring the political leaning or slant of many different types of content at scale, we instead varied exposure to content based on the estimated political leaning of the source of the information. Using a Facebook classifier, we estimate the political leaning of other users directly (see SI Section S1.3 for details). Building on prior research (14,15,21,33,34), we estimate the political leanings of Pages and Groups using the political leanings of their audience (e.g., Group members and Page followers). For participants assigned to treatment, we downranked all content (including, but not limited to, civic and news content) from friends, Groups, and Pages who were predicted to share the participant's political leaning using this approach (e.g., all content from conservative friends as well as Groups and Pages with conservative audiences was downranked for participants classified as conservative; see SI Section S1.1).

We also added the following passage to the limitations noting this distinction and identifying "content-based measures of echo chambers" as a direction for future research:

Fourth, our study examines the prevalence of "echo chambers" using the estimated political leanings of users, Pages, and Groups who share content on social networks. We do not directly measure the slant of the content that is shared; doing so would be a valuable contribution for future research.

7. I may have missed this, but to what population are the results of the experiment meant to generalize via the weights – is it American Facebook users or American voters? It could be worthwhile to report both weighted estimates.

The survey weights we employed were created to reflect the population of adult monthly active Facebook users who were eligible for recruitment, not American voters (as stated in the Design section; see SI Section S10 for details). We did not construct weights to generalize our results to other populations and thus are unfortunately unable to provide a separate set of weighted estimates for these quantities. Weighting to voters in particular raises extrapolation concerns (the population includes many non-Facebook users) as well as data issues (even if we tried to weight to the population of Facebook users who vote, the data used to create our population targets did not include a measure of validated voting). Ultimately, our goal is to estimate effects that generalize to the broadest possible set of users.

In conclusion, we thank the editor and the reviewers for your close reading and feedback and hope that you will see the paper as ready for publication in *Nature*!

References

David Broockman. "Approaches to studying policy representation." *Legislative Studies Quarterly* 41(1): 181-215, 2016

Nir Grinberg, Kenneth Joseph, Lisa Friedland, Briony Swire-Thompson, and David Lazer. "Fake news on Twitter during the 2016 U.S. presidential election." *Science* 363(6425): 374-378, 25 Jan 2019

Andrew M. Guess. "(Almost) everything in moderation: New evidence on americans' online media diets." *American Journal of Political Science* 12589, 2021.

Magdalena Wojcieszak, Sijfra de Leeuw, Ericka Menchen-Trevino, Seungsu Lee, Ke Maddie Huang-Isherwood, and Brian Weeks. "No polarization from partisan news." *International Journal of Press and Politics*, Sep 2021

Reviewer Reports on the First Revision:

Referee #1 (Remarks to the Author):

I believe the revisions adequately address my questions and suggestions.

Referee #2 (Remarks to the Author):

The authors addressed my comments and suggestions and made several improvements to the article that clarify the results and their broader context. The added great detail about the ideology classifiers provides impressive results and a strong basis for their use in the observational analysis and the experiment. They also added more information that I requested on the role of Meta's Legal/Privacy department in the writing of the report and on some of the estimates. Their response letter also helped me to understand some more details about the effect of the treatment and sample sizes. While the reporting of the full sample size is still vague and still in relation to the total FB population, I think this is acceptable given the large scale of the analysis and the sensitive nature of such statistics for a social networking platform. While the revision improved a lot on some of my comments, there are two I still have to mention:

- Thanks a lot for the clarification about the effect of the treatment, especially the new text about cross-cutting sources and how that was not a direct target of the intervention and the extra figure in the SI about it. Here, I still would like to stress that the magnitude of the effect of the intervention appears only on page three, while the abstract describes it as "we substantially reduced exposure to content from like-minded sources during the 2020 U.S. election". As a reader, I had a much higher expectation of the extent of the effect until I got to understand it from the main text, so a simple solution here would be to disclose in the abstract that the reduced exposure is not "substantial", but "about a third", as it is described later. I think this is an important point to reflect clearly everywhere in the paper since one of the most important take-home messages is the limited power of an algorithm change intervention that does not severely alter other aspects of the platform.

- I appreciate the explanations in the reply about the interpretation of the 20.6% of users with high like-minded exposure. Authors rightfully point to the subjective nature of the interpretation of this statistic, which depends on the prior expectations about the prevalence of echo chambers. Some of us might not expect this number to be much higher but a general audience, even among scientists, might overestimate that value. The text added to the conclusions is a good step, but I still fear that the reporting of the 20.6% fraction on the results needs more information. On the results, I would rather add more transparency to the reporting of statistics and leave it more to the readers to conclude whether this is rare, infrequent, or any other subjective term. For example, beyond the 20.6% fraction, I think the authors should also report the numerator and denominator of that fraction for readers to have all information and come to their own conclusions in a transparent way when reading the authors' discussion section.

As a final remark, I'd like to remind the authors and editor of my high impression of this article, its robust and careful analysis, and the important role it can have in future policy discussions. Do not take my focus on the two above points as a negative evaluation of the article, I still think this is a great piece for Nature and I only want to help by trying to clarify those comments.

Referee #3 (Remarks to the Author):

In their revision of "Like-minded sources on Facebook: Prevalent but not polarizing," the authors address many of the comments that I made in my previous review. I appreciate the time and

effort that the authors committed to addressing these issues, even performing an additional analysis that looks for possible heterogeneous effects by age.

The observational and experimental studies that the authors conducted remain novel and interesting, and I believe they will be an important addition to the public discussion of echo chambers and polarization. My remaining critiques still have to do with the presentation and framing of the results, which I find to be unnecessarily skewed in favor of arguing that the intervention did not “work.” For example, the authors find that the intervention reduced exposure to incivility and misinformation – while the strict benefits of the former are debated (e.g., Rossini 2019, Edyvane 2020), those of the latter are not. Yet both of these effects are entirely left out of the Discussion section, while the authors instead focus on how engagement “rates” with like-minded sources increased due to the intervention. While it is certainly interesting that users still seek out like-minded sources when they are made less accessible, this focus seems to downplay the fact that “total” engagement with like-minded content did decrease (and incivility and misinformation, in turn).

In the final paragraphs of the Introduction to the manuscript, the authors introduce the survey results with “Most importantly.” It is not immediately clear to me why the survey results are more important than the persistent effects on misinformation exposure. If users see one fewer pieces of misinformation every two days, is that not as important as a response to a survey item? I do not have the answer to this question, of course, and in some sense it is bigger than this project. But it seems like it should be important to consider each set of results in their own right, without prioritizing one over the other, and to consider the effects of the intervention as complicated and not uniformly good or bad (or neutral).

Minor comment: I find the concluding sentence of the paper to be a bit odd, given the focus on ‘political identity’ in light of the authors findings that politics makes up a relatively small portion of what people see on social media (a finding that I still would have liked to see centered more in the concluding paragraphs). Perhaps the *political* information we see on social media is a reflection of our political identity, but the information we see more generally is, arguably, disconnected from political identity.

Author Rebuttals to First Revision:

R2's comments:

Thanks a lot for the clarification about the effect of the treatment, especially the new text about cross-cutting sources and how that was not a direct target of the intervention and the extra figure in the SI about it. Here, I still would like to stress that the magnitude of the effect of the intervention appears only on page three, while the abstract describes it as "we substantially reduced exposure to content from like-minded sources during the 2020 U.S. election". As a reader, I had a much higher expectation of the extent of the effect until I got to understand it from the main text, so a simple solution here would be to disclose in the abstract that the reduced exposure is not "substantial", but "about a third", as it is described later. I think this is an important point to reflect clearly everywhere in the paper since one of the most important take-home messages is the limited power of an algorithm change intervention that does not severely alter other aspects of the platform.

I appreciate the explanations in the reply about the interpretation of the 20.6% of users with high like-minded exposure. Authors rightfully point to the subjective nature of the interpretation of this statistic, which depends on the prior expectations about the prevalence of echo chambers. Some of us might not expect this number to be much higher but a general audience, even among scientists, might overestimate that value. The text added to the conclusions is a good step, but I still fear that the reporting of the 20.6% fraction on the results needs more information. On the results, I would rather add more transparency to the reporting of statistics and leave it more to the readers to conclude whether this is rare, infrequent, or any other subjective term. For example, beyond the 20.6% fraction, I think the authors should also report the numerator and denominator of that fraction for readers to have all information and come to their own conclusions in a transparent way when reading the authors' discussion section.

We made the requested change in the abstract and replaced the word "substantial" with a more precise description of the magnitude of the decrease in exposure. To comply with the guidelines of *Nature*, which specify that no numerical information should be presented in the abstract, we state per the reviewer that exposure decreased by "about one-third."

The reviewer asks for more information about the statistic that 20.6% of users have frequent exposure to content from like-minded sources. Extended Data Table 1 (previously Table S14 in the Appendix) provides and contextualizes this number, and we now refer to Extended Data

Table 1 in the paragraph in which we present that statistic; the table itself makes clear what the denominators are.

R3's comments:

The observational and experimental studies that the authors conducted remain novel and interesting, and I believe they will be an important addition to the public discussion of echo chambers and polarization.

My remaining critiques still have to do with the presentation and framing of the results, which I find to be unnecessarily skewed in favor of arguing that the intervention did not “work.” For example, the authors find that the intervention reduced exposure to incivility and misinformation – while the strict benefits of the former are debated (e.g., Rossini 2019, Edyvane 2020), those of the latter are not. Yet both of these effects are entirely left out of the Discussion section, while the authors instead focus on how engagement “rates” with like-minded sources increased due to the intervention. While it is certainly interesting that users still seek out like-minded sources when they are made less accessible, this focus seems to downplay the fact that “total” engagement with like-minded content did decrease (and incivility and misinformation, in turn).

We now more clearly convey that our experimental intervention reduced exposure to incivility and misinformation in the discussion section. We write:

Our field experiment also shows that changes to social media algorithms can have dramatic effects on the content users see. The intervention substantially reduced exposure to content from like-minded sources, which also had the effect of reducing exposure to content classified as uncivil and content from sources that repeatedly post misinformation. However, the tested changes to social media algorithms cannot fully counteract people’s proclivity to seek out and engage with congenial information. Consenting participants who were randomized to see less content from like-minded sources were actually *more* likely to engage with such content when they encountered it.

In the final paragraphs of the Introduction to the manuscript, the authors introduce the survey results with “Most importantly.” It is not immediately clear to me why the survey results are more important than the persistent effects on misinformation exposure. If users see one fewer pieces of misinformation every two days, is that not as important as a response to a survey item? I do not have the answer to this question, of course, and in some sense it is bigger than this project. But it seems like it should be important to consider each set of results in their own right, without prioritizing one over the other, and to consider the effects of the intervention as complicated and not uniformly good or bad (or neutral).

We delete “Most importantly” from the paragraph and replace it with “Furthermore.” Regarding the reviewer’s comment about misinformation exposure, we note that, as pre-registered, the primary goal of the experiment was to test the effects of the intervention on attitudes. As such,

while we do show the effects of the treatment on misinformation exposure in our results and mention them in the discussion, they are not a central focus of the paper.

*Minor comment: I find the concluding sentence of the paper to be a bit odd, given the focus on 'political identity' in light of the authors findings that politics makes up a relatively small portion of what people see on social media (a finding that I still would have liked to see centered more in the concluding paragraphs). Perhaps the *political* information we see on social media is a reflection of our political identity, but the information we see more generally is, arguably, disconnected from political identity.*

The reviewer is correct that we cannot say with certainty the extent to which *political* identity shapes the content people choose to see. However, given the results of our experiment and the scope of our intervention, we do feel confident in asserting that for most people, the information they see on social media is a reflection of their identities (political or otherwise) rather than a major *cause* of the attitudes associated with those identities. Thus, we removed the word “political” from that final sentence:

“The information we see on social media may be more a reflection of our identity than a source of the views we express.”